# Physical neural networks using sharpness-aware training

**Tengji Xu[1,2], Zeyu Luo[1,2], Shaojie Liu [1,2], Li Fan[1,2], Qiarong Xiao[1], Benshan Wang [1], Dongliang Wang [1] & Chaoran Huang [1]** ✉

Recent advances in AI are pushing the limits of traditional hardware, making physical neural networks (PNNs) a promising alternative. However, training PNNs remains challenging: in silico training suffers from model-reality mismatch, while in situ training produces device-specific models that do not transfer across fabrication variations. Both approaches are further compromised by post-deployment perturbations, such as thermal drift or misalignment, which invalidate trained models and require retraining. We address these challenges through sharpness-aware training (SAT), inspired by sharpness-aware minimization, which links loss landscape geometry to generalization. We establish a connection between loss landscape sharpness and robustness in physical systems and leverage it to improve PNN training. SAT is compatible with both in silico and in situ settings: it mitigates model-reality gaps, enables cross-device transfer, and provides strong resilience to post-deployment perturbations without retraining. We demonstrate SAT across three PNN platforms and multiple tasks, including classification, compression, reconstruction, and generation, showing its broad applicability.

The tremendous success of modern artificial intelligence (AI) is driven by a synergic operation of hardware and algorithms[1]. Powerful hardware, particularly graphics processing units, has revolutionized neural network (NN) computing by drastically reducing the time and resources needed to process large datasets. Central to this progress is backpropagation (BP), the foundational algorithm for training NNs, which efficiently dictates how NNs learn and evolve by minimizing errors[2].

Recent AI models have begun to strain the capabilities of traditional digital hardware. The cost of training and inferring AI models doubles every 2 months, outpacing Moore's Law. Consequently, AI systems are increasingly facing challenges caused by hardware efficiency. To address hardware limitations, a different computing paradigm, neuromorphic computing, has gained traction. Many neuromorphic computers are physical systems leveraging photonics[3–5], analog electronics[6], and other wave-based physics[7,8], as shown in Fig. 1a, b. The physical models of these systems exhibit mathematical isomorphism with NNs, enabling their computing

capabilities. Their physical configurations can alleviate the data movement bottleneck found in traditional digital hardware, substantially improving both computing speed and energy efficiency. Large-scale[9–13], high-speed[14–17], and energy-efficient[18–22] physical neural networks (PNNs) have been widely reported[23–25].

However, the development of suitable training methods that are synergistic with these computing hardware has lagged behind. The training process remains cumbersome and unreliable for PNNs. Traditional gradient-based algorithms, such as BP, cannot be directly applied to PNNs due to the difficulty of obtaining gradients from physical systems. Moreover, even with appropriate training methods, PNNs struggle to maintain computing accuracy during deployment, as physical systems are inherently highly susceptible to disturbances compared to digital hardware, leading to errors[3,26,27].

In reviewing the existing training methods, they can be divided into two main categories: in silico training and in situ training (Fig. 1c). To train a PNN in silico, the process begins by extracting a digital model of the physical system. The training is then conducted on a

[1]Department of Electronic Engineering, The Chinese University of Hong Kong, Hong Kong SAR, China. [2]These authors contributed equally: Tengji Xu, Zeyu Luo, Shaojie Liu, Li Fan. ✉e-mail: crhuang@ee.cuhk.edu.hk

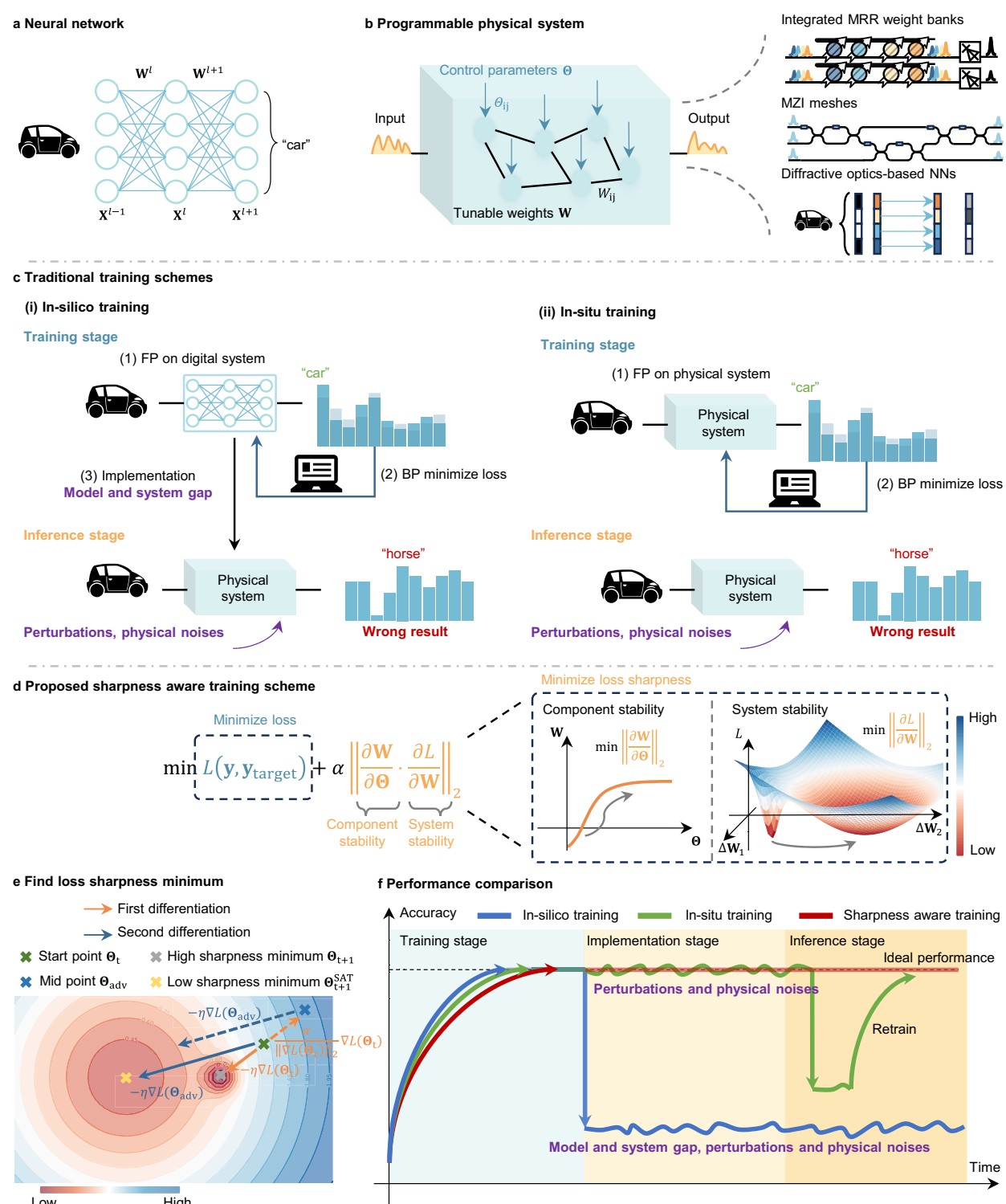

**Fig. 1 | The detailed comparison of physical neural network training methods and the proposed sharpness-aware training method's principle. a** Diagram illustrating a typical neural network. The neural network contains synapses to perform matrix-vector multiplications followed by nonlinear activation functions. **b** Schematic diagram of a programmable physical system with control parameters **Θ** and tunable weights **W**. The tunable weights **W** are directly controlled by the control parameters **Θ**. The system takes input signals and produces output signals based on the adjusted parameters. **c** Illustration of (i) in silico training and (ii) in situ training methods. The tunable weights are achieved by separately controlling different physical parameters, including currents and phases. **d** Proposed sharpness-aware training scheme. The training goals include reducing loss while increasing component and system stability, respectively. **e** Schematic diagram of the proposed sharpness-aware training scheme parameter update. **f** Performance comparison between in silico training, in situ training, and sharpness-aware training. MRR microring resonator, MZI Mach-Zehnder interferometer, NN neural network, FP forward propagation, BP backpropagation, adv adversary, SAT sharpness-aware training.

digital computer, typically using BP. Once the model is trained, the parameters are applied to the PNN. However, in silico training requires highly accurate digital modeling of the physical system, which is challenging due to unavoidable fabrication-induced errors and intrinsic noise present in the physical systems. As a result of such a model-reality gap, the inference accuracy is often lower than the ideal value, as illustrated in Fig. 1c-i. In contrast, in situ training alleviates the need for precise digital modeling by looping the physical system directly into the training process. However, in situ training faces the problem of obtaining the gradients of the physical system. Current methods for obtaining gradients include: (1) Approximating the gradients using a digital twin model of the physical system obtained through data-driven methods[4,28–31]; (2) Estimating the gradients through finite-difference methods by applying perturbations and measuring difference[32–34]; (3) Directly measuring the gradients physically[35–40]. Each of these approaches introduces overhead, either through excessive measurement requirements or the computational cost of creating a digital twin model. To mitigate this overhead, various BP-free or gradient-free approaches have been proposed[3,41,42]. However, their training efficiency and effectiveness in complex applications have yet to be fully demonstrated. Moreover, a drawback of all in situ learning approaches is that, because the specific physical system is looped into the training process, the trained results are only applicable to that particular system and cannot be transferred to other systems, even those with the same design. Our simulation results show that the in situ-trained model is even more vulnerable to imperfections than the in silico-trained model when transferring the model to other systems. A critical issue across both in silico and in situ training methods is that perturbations after deployment, such as thermal drift or alignment errors, will make the trained parameters ineffective, leading to accuracy loss and requiring retraining[3].

This work proposes and demonstrates a training technique, sharpness-aware training (SAT), that overcomes the challenges of both in silico and in situ training. SAT can be implemented in both in silico and in situ training settings. It closes the model-reality gap in in silico training by enabling accurate training of PNNs using efficient BP algorithms, without the need for an exact digital model or looping the actual hardware during the process. The systems trained in silico by SAT even outperform those trained in situ, even in the presence of modeling and fabrication errors. Additionally, SAT addresses a key limitation of current in situ training methods. Parameters trained on one device can be reliably transferred to other devices for inference without accuracy loss, even if they are different due to fabrication variances. Equally important, the technique exhibits strong resilience to post-deployment perturbations, a common problem in both in silico and in situ training. This robustness allows PNNs to continue operating accurately under perturbations without the need for retraining.

The key to these benefits is that SAT leverages the geometry of the loss landscape in a physical system: it minimizes not only the loss value but also the loss sharpness, which indicates the robust region of the physical system, as shown in Fig. 1d. By identifying parameters in regions of uniformly low loss, SAT ensures that the loss remains minimal despite challenges common in analog computing systems, such as modeling inaccuracies, fabrication variances, and external perturbations. SAT is inspired by the machine learning technique sharpness-aware minimization (SAM)[43], which enhances model generalization by connecting loss sharpness to model generalization. In contrast, SAT introduces a different connection between loss sharpness and the robustness of physical systems. Furthermore, SAT develops methodologies for automatically locating flat minima in physical systems and enables direct training in physical systems.

SAT is universally applicable to various PNNs, regardless of whether their models are explicitly known or unknown. We demonstrate the generality of SAT through three typical photonic NNs: integrated microring resonator (MRR) weight banks, Mach-Zehnder

interferometer (MZI) meshes, and diffractive-optics-based NNs. These demonstrations highlight the effectiveness and accuracy of SAT during both training and deployment stages, even under error sources inherent in analog computing. This includes imprecise modeling prior training, thermal and electrical disturbances that heavily affect sensitive devices such as resonators, fabrication variances in integrated optics, and deployment errors commonly seen in free-space optical systems. We further expand the validation across different tasks, including image classification, compression, reconstruction, and generation. Overall, SAT offers a practical, effective, and computationally efficient solution for training and deploying PNNs in real-world applications.

## Results
### Sharpness-aware training: seeking loss minimum and loss sharpness minimum in a physical system
A deep neural network (DNN) is a multi-layer system, where each layer consists of synapses and nonlinear nodes, as depicted in Fig. 1a. The mathematical model of the $l$th layer of DNN is given by,

$$\mathbf{X}^{l+1} = \mathbf{A}^l(\mathbf{W}^l\mathbf{X}^l + \mathbf{b}^l) \tag{1}$$

Here, $\mathbf{X}^l$ is the input of the $l$th layer, $\mathbf{W}^l$ and $\mathbf{b}^l$ denote the trainable weights and bias, and $\mathbf{A}^l(\cdot)$ represents the $l$th layer nonlinear activation layer. A PNN emulates this process through a physical system controlled by a set of tunable physical parameters $\mathbf{\Theta}$, such as electrical currents. These parameters determine the system's transfer function, which can be expressed by $y = F(x; \mathbf{\Theta})$ with x and y representing the system input and output and $\mathbf{\Theta}$ representing the tunable physical parameters. Tuning these control parameters affects the system's transfer function, making them equivalent to changing weights in a DNN. Therefore, training a PNN is to find the optimal control parameters that minimize the loss function for a given task.

To train such a system accurately, obtaining the exact mathematical expression for $y = F(x; \mathbf{\Theta})$ is essential for conventional methods. However, this is nearly impossible for real-world physical systems, which are governed by highly complex physics and are prone to parameter drift, device crosstalk, and environmental disturbances. Discrepancies between the ideal model and the actual system lead to training errors. In situ training reduces the reliance on an exact mathematical expression. It obtains the true loss function by experimental measurement. With the true loss function, even if there are slight discrepancies between the hardware and the ideal model, the loss can still be minimized through iterative optimization[4]. However, in situ training is inefficient because it is specific to the particular system being trained and cannot be easily transferred to other devices, even those fabricated with the same design. This limitation is unfavorable compared to the typical functioning of modern AI hardware, where training is conducted on a single high-end computer or a cloud-based computer cluster, and the trained parameters are then deployed across many edge devices for inference[26]. Moreover, a critical issue across both in silico and in situ training methods is that perturbations after deployment, such as thermal drift or installation errors, can render the trained parameters invalid.

We address all the challenges of both in silico and in situ training with our SAT method, which leverages the geometry of the loss landscape in a physical system to modify the training objective accordingly. The loss landscape, illustrated in Fig. 1e, represents how the loss changes as parameters are adjusted. Traditional training methods like BP only focus on minimizing the loss function, which typically ends at the "High sharpness minimum" as depicted in Fig. 1e. The "High sharpness minimum" is characterized by sharp curvatures, where small changes in parameters lead to rapid increases in loss. In machine learning, such suboptimal minima lead to poor model generalization. In this work, we associate these sharp minima in the loss landscape

with the poor robustness of physical systems. At such sharp minima, slight parameter changes due to modeling discrepancies, fabrication variances, or environmental disturbances can cause rapid increases in loss and errors, affecting both the training and deployment stages when using in silico and in situ training methods.

Our SAT method addresses these problems by automatically searching for flat minima in the loss landscape, as the "Low sharpness minimum" illustrated in Fig. 1e. At these flat minima, the loss remains minimal even with parameter changes, providing high resilience and robustness against errors and imperfections in physical systems. In this work, we derive a universal framework for finding flat minima across different PNN systems.

To find the "Low sharpness minimum," the goal is not only to minimize the loss but also to ensure parameters lie in the neighborhoods with uniformly low loss value. The optimization process of SAT can be mathematically expressed as finding $\mathbf{\Theta}$ minimizing the loss function $L_1$, as shown in Eq. (2):

$$
\begin{aligned}
L_1 &= L(y, y_{\text{target}}; \mathbf{\Theta}) + \alpha \left\| \frac{\partial L(y, y_{\text{target}}; \mathbf{\Theta})}{\partial \mathbf{\Theta}} \right\|_2 \\
&= L(y, y_{\text{target}}; \mathbf{\Theta}) + \alpha \left\| \underbrace{\frac{\partial \mathbf{W}}{\partial \mathbf{\Theta}}}_{\text{Component stability}} \cdot \underbrace{\frac{\partial L(y, y_{\text{target}}; \mathbf{\Theta})}{\partial \mathbf{W}}}_{\text{System stability}} \right\|_2
\end{aligned}
\tag{2}
$$

The first term, $L(y, y_{\text{target}}; \mathbf{\Theta})$, in the loss function seeks to find the loss minima, while the second term, $\left\| \frac{\partial L(y, y_{\text{target}}; \mathbf{\Theta})}{\partial \mathbf{\Theta}} \right\|_2$ reflects the sensitivity of the objective function to changes in the parameters. This term aims to identify parameters that lie in the neighborhoods with uniformly low loss values. As shown in Fig. 1d, this additional term $\frac{\partial L(y, y_{\text{target}}; \mathbf{\Theta})}{\partial \mathbf{\Theta}}$ can be expressed as $\frac{\partial L(y, y_{\text{target}}; \mathbf{\Theta})}{\partial \mathbf{W}} \cdot \frac{\partial \mathbf{W}}{\partial \mathbf{\Theta}}$, where $\frac{\partial \mathbf{W}}{\partial \mathbf{\Theta}}$ can optimize the stability of individual components and $\frac{\partial L(y, y_{\text{target}}; \mathbf{\Theta})}{\partial \mathbf{W}}$ optimizes the stability of the whole system. Here, the component stability refers to how the weight of a single component remains stable under small control parameter perturbations, while the system stability reflects how the final loss remains stable under small weight perturbations. Both component and system stability are improved during optimization. The regularization term $\alpha$ determines the penalty of sharpness for loss during the actual training process, that is, the trade-off between sharpness and loss.

However, directly calculating the second term requires computing second-order derivatives, the Hessian matrix, which leads to the computational complexity increasing from $O(n)$ to $O(n^2)$, where $n$ denotes the matrix size. This additional cost makes Eq. (2) challenging to implement in practice. To address this, Foret et al. propose approximating the gradient for minimizing the loss function using a first-order gradient of $L$ at $\mathbf{\Theta} + \Delta\mathbf{\Theta}$[43], the gradient is calculated through Eq. (3),

$$
\begin{aligned}
\Delta\mathbf{\Theta} &= \alpha \frac{\partial L/\partial \mathbf{\Theta}}{\| \partial L/\partial \mathbf{\Theta} \|_2} \\
\frac{\partial L_1(\mathbf{\Theta})}{\partial \mathbf{\Theta}} &= \left. \frac{\partial L(\mathbf{\Theta} + \Delta\mathbf{\Theta})}{\partial \mathbf{\Theta}} \right|_{\Delta\mathbf{\Theta} = \alpha \frac{\partial L/\partial \mathbf{\Theta}}{\|\partial L/\partial \mathbf{\Theta}\|_2}}
\end{aligned}
\tag{3}
$$

Here, $\Delta\mathbf{\Theta}$ represents the parameter perturbation that induces the largest change in the loss within the neighborhood of $\mathbf{\Theta}$. The previously defined hyperparameter $\alpha$ also indicates how large the perturbation $\Delta\mathbf{\Theta}$ should be given to $\mathbf{\Theta}$. And $\partial L/\partial \mathbf{\Theta}$ depicts the loss gradient with respect to parameter $\mathbf{\Theta}$. The specific value of $\alpha$ is chosen based on whether it can maximize the model's robustness without decreasing the accuracy. We give a sensitivity analysis over the hyperparameter $\alpha$, and the optimum is around 0.1 to 1. The detailed sensitivity analysis is

depicted in the Supplementary Note 1.2. According to the result, we choose the value of $\alpha$ to be 0.1.

Minimizing the loss function can thus be achieved through a two-step process of automatic differentiation, as illustrated in Fig. 1e. The first step involves identifying the point $\mathbf{\Theta} + \Delta\mathbf{\Theta}$ that results in the maximum loss within the neighborhood (denoted as "Mid point" in Fig. 1e) using Eq. (3). Once these parameters are identified, the gradient is recalculated at "Mid point." This second gradient calculation with Eq. (3) reflects the direction in which the maximum loss in the neighborhood can be reduced, i.e., directing to the "flat minimum," as shown at the point labeled "Low sharpness minimum" in Fig. 1e. Consequently, this two-step BP not only finds the minimum loss but also ensures that the loss remains low in the vicinity of the minimum, achieving a "flat minimum."

The loss minimization process highlights the difference from our prior work, optical pruning[27]. Optical pruning focuses solely on improving the stability of individual devices inside the PNNs. In contrast, SAT not only improves the stability of individual components but also enhances the overall system's stability by ensuring that the entire system's loss function remains minimized in response to weight perturbations. Furthermore, optical pruning is limited to relatively simple systems where such stable regions could be easily identified. For instance, in microring circuits, where each component operates almost independently, the robust region corresponds to the flat region of each individual microring's transfer function. In contrast, SAT is broadly applicable to different physical systems because it can automatically identify robust minima without requiring explicit knowledge of the underlying physical dynamics. SAT can also outperform noise-aware training (NAT)[44–47], a widely used method to improve physical system robustness by injecting synthetic noise during training. The reason is that NAT typically injects simple additive Gaussian noise into the model, but real-world imperfections, such as modeling inaccuracies, pixel-level misalignments, fabrication errors, and temperature-induced shifts, often deviate from this assumption. Consequently, NAT struggles to fully capture the complex, system-specific non-Gaussian error characteristics, limiting its effectiveness in enhancing robustness. In contrast, SAT makes no assumptions about the nature of system imperfections. Instead, by connecting robustness to the intrinsic geometry of the loss landscape, SAT formulates a general optimization objective and naturally accounts for diverse and non-Gaussian imperfections without requiring explicit modeling. This allows SAT to optimize PNNs with greater accuracy and resilience, even when the specific forms of system noise are unknown and non-Gaussian.

As shown in Fig. 1f, SAT is applicable to both mainstream in silico and in situ training workflows. In in silico training, the training process is conducted on the mathematical model y = $F$(x; $\mathbf{\Theta}$) of a PNN to directly determine the controlling parameters. Even when the mathematical model deviates from the actual system, the trained parameters remain valid and maintain high accuracy when applied to the physical system. In in situ training, the loss function is calculated directly from measurements, and the two-step differentiations are obtained using the gradient estimation methods mentioned earlier. We further demonstrate that, compared to conventional in situ training, SAT not only achieves the same accuracy on the specific physical device being trained but also enables the trained parameters to be transferable to other devices. This contrasts with other in situ training methods.

In the following section, we demonstrate SAT on three distinct PNNs. These demonstrations not only highlight the broad applicability of SAT, but also illustrate its advantages over existing training methods from three perspectives: (1) When SAT is implemented in in silico training setting, it closes the model-reality gap, enabling accurate BP training under imprecise models (section "In silico SAT: enabling accurate training with an inaccurate digital model"); (2) When SAT is implemented in in situ training setting, SAT addresses the challenge of

limited transferability caused by fabrication variances. Parameters trained on one device using SAT can be reliably deployed to other devices without accuracy degradation, even when hardware discrepancies exist (section "In situ SAT: facilitating transferable in situ training"); (3) In both in silico and in situ training settings, PNNs trained by SAT can continuously operate accurately after training under perturbations without retraining (sections "In silico SAT: enabling accurate training with an inaccurate digital model", "In situ SAT: facilitating transferable in situ training," and "Extending SAT to PNNs without explicitly known models"); (4) SAT is broadly applicable to physical systems regardless of whether an accurate physical model is explicitly available (sections "In silico SAT: enabling accurate training with an inaccurate digital model", "In situ SAT: facilitating transferable in situ training," and "Extending SAT to PNNs without explicitly known models"). The advantages of SAT over existing in silico and in situ training methods, including those designed to improve robustness, such as NAT and optical pruning, are further demonstrated in the following sections, and summarized in Table 1.

## In silico SAT: enabling accurate training with an inaccurate digital model

This section demonstrates our approach's effectiveness in the in silico training scheme. Using MRR-based PNNs as our first demonstration platform, we demonstrate two advantages of SAT compared to traditional training techniques. Firstly, SAT closes the model-reality gap: it does not require highly accurate digital modeling of the physical system or considering side effects or system imperfections, yet it still outperforms the method requiring them. Secondly, PNN trained by SAT is more robust against dynamic noise and post-deployment perturbations, particularly temperature drift.

In the MRR-based PNNs, weight synapses are realized using MRR weight banks, as illustrated in Fig. 2a. The input matrix is represented by an array of analog signals, each modulated onto a laser at a distinct wavelength. The multiplication operation is performed by an array of MRR weight banks, shown in Fig. 2b. Each MRR in the weight bank has a slightly offset radius, resulting in a unique resonance wavelength that aligns with the corresponding input laser's wavelength. By tuning the resonance wavelength of the MRR, the distribution of light between the Drop and Through ports can be precisely controlled, allowing for tunable weights, as shown in Fig. 2c. Thermal tuning is the commonly employed mechanism for controlling MRRs due to its ease of implementation on silicon photonics chips. Once the input signals are weighted, they are detected by a balanced photodetector (PD)[17,27]. In this experiment, we use a single-end PD to detect the light from the Through port. MRR-based PNNs offer several advantages, including power-efficient tuning, straightforward weight assignment, and high computational density due to the compact device footprint[24]. However, MRRs are highly sensitive to thermal changes, such as thermal drift, thermal crosstalk, and ambient disturbances. In our previous work[27], we observed that a resonance shift of just 12 pm (corresponding to a temperature change of 0.2 °C) caused the PNN's accuracy to drop from an ideal 99.0% to 67.0% when classifying the Modified National Institute of Standards and Technology (MNIST) dataset[48].

Here, we conduct our experiment using a 4 × 4 MRR weight bank, as shown in Fig. 2b (refer to the Supplementary Notes 4.2 and 4.3 for detailed device design and experimental process). To demonstrate the effectiveness of SAT to address challenging problems, we have several experimental settings. (1) When constructing the mathematical model $y = F(x; \Theta)$, we treat all MRRs as identical. In this model, $\Theta$ indicates an array of currents used to control each MRR. We experimentally measure the current-weight relationship on only one specific MRR and apply this relationship uniformly across all MRRs in the model. However, as shown in Fig. 2d, fabrication variations lead to different current-weight characteristics for each MRR, resulting in modeling

errors. (2) We intentionally ignore all side effects that could introduce modeling errors in our training model. These side effects include thermal crosstalk between devices, noise, and imperfections in peripheral circuits such as wavelength-dependent gain in erbium-doped fiber amplifier and non-ideal modulations. Ignoring these factors further causes the mathematical model to deviate from the actual system. (3) After training, the deployed system experiences thermal drift, further impacting the system's performance.

We first compare the training performances of our SAT method with the traditional training method using standard BP on the MNIST database. As shown in Fig. 2e, both methods converge after 20 training epochs with a training accuracy of 99.0%. (see Supplementary Note 4.1 for training details). The current distributions obtained using the two methods are shown in Fig. 2f. The effectiveness of SAT is highlighted by the loss landscape illustrated in Fig. 2g. The loss landscape clearly shows that the system trained with SAT exhibits a flatter profile, indicating enhanced stability and robustness against variations in control parameters compared to traditional training methods[49]. To emphasize the largest curvatures, the x and y axes represent the two principal directions of the Hessian matrix of the loss function with respect to the control currents, while the z-axis corresponds to the loss function. This robustness is also reflected in the maximum eigenvalue of the Hessian matrix, $\lambda_{max}$. A smaller eigenvalue suggests a flatter loss landscape. With SAT, $\lambda_{max}$ reduces dramatically from 746.8 with standard BP to just 1.2, highlighting the effectiveness of SAT in identifying stable regions within the physical system.

After training, we deploy the trained currents to the MRR-based PNN to evaluate the inference accuracy. As mentioned in the experimental setup, there is a deviation between the model used for training and the actual system due to static errors and dynamic noise. Consequently, when using standard BP, the accuracy for MNIST classification drops to 80.0% from the ideal accuracy of 98.0%. In comparison, the accuracy achieved with SAT remains at 97.0%, as shown in Fig. 2h. This result is obtained at a temperature of 22 °C, the same temperature used to characterize the MRR current-weight relationship. To further simulate the real-world conditions under temperature drift, we vary the chip temperature using a thermoelectric controller (TEC) from 21 to 23 °C and test the inference accuracies, with results shown in Fig. 2h. The system trained with SAT demonstrates remarkable robustness to temperature changes, maintaining its computational accuracy, while the accuracy of the system trained with standard BP degrades from 80.0 to 17.0% as the temperature varies from 21 to 23 °C.

Next, we directly remove the TEC and measure the accuracy under the room temperature of 20 °C. Note that the training model is still characterized at 22 °C with TEC. In this circumstance, despite the temperature change being as large as 2 °C, SAT still maintains high accuracy at 91.0%, in contrast to 52.0% for standard BP.

We further compare SAT with NAT. NAT achieves an accuracy of 73.0%, indicating that it is less resilient to such a temperature variation compared to SAT, which achieves 91.0%. This difference is also evident in the sharpness metric $\lambda_{max}$: while NAT reduces $\lambda_{max}$ compared to standard BP, it still reaches 414.5, which remains substantially higher than 1.2 achieved by SAT. The reason is that NAT typically only accounts for Gaussian errors but struggles to fully capture the complex, system-specific non-Gaussian error characteristics, limiting its effectiveness in enhancing robustness. In contrast, SAT naturally accounts for diverse and non-Gaussian, allowing SAT to optimize PNNs with greater accuracy and resilience, even when the specific forms of system noise are unknown and non-Gaussian. We also compare the SAT method with our previous work on optical pruning[27]. The core idea in that work is to train the parameters of the MRR-based PNN to enhance the noise robustness of each MRR. Optical pruning focuses solely on improving the stability of individual devices. Under large model deviations and temperature fluctuations, optical pruning achieves an accuracy of only 73.0%. In contrast, SAT not only improves

**Table 1 | Comparison of training methods regarding training objectives, generality, and performance during the training, deployment, and inference stages**

| Training method | | Training objective | Generality | Training stage | | | | Deployment stage | Inference stage | |
|---|---|---|---|---|---|---|---|---|---|---|
| | | | | Tolerance to the model-reality gap | Training speed | Robustness | Scalability | Accuracy | Accuracy (under ambient perturbations) | Model transferability |
| In silico training | | Loss | High ✓ | Low | $O(T_0)$ ✓ | Low | High ✓ | Low | Low | Mid |
| In situ training | Gradient approx. by data-driven (PAT[4], DAT[28]) | Loss | High ✓ | Mid | $O(T_0)$ ✓ | Low | High ✓ | High ✓ | Mid | Low |
| In situ training | Gradient estimation (finite difference[32–34]) | Loss | High ✓ | High ✓ | $>O(T_0)$ | Low | Mid | High ✓ | Mid | Low |
| In situ training | Gradient measurement (fully forward[35], Adjoint)[36,38] | Loss | Limited (Require symmetry) | High ✓ | $O(T_0)$ ✓ | Low | High ✓ | High ✓ | Mid | Low |
| In situ training | Physical local learning (forward forward[3]) | Loss | High ✓ | High ✓ | $>O(T_0)$ | Low | Mid | High ✓ | Mid | Low |
| In situ training | Direct feedback alignment[41,42] | Loss | High ✓ | High ✓ | $>O(T_0)$ | Low | Mid | High ✓ | Mid | Low |
| In situ training | Gradient-free methods (GA, SO, ES) | Loss | High ✓ | High ✓ | $>>O(T_0)$ | Low | Mid | High ✓ | Mid | Low |
| In silico NAT | | Loss | High ✓ | Mid (Gaussian errors only) | $O(T_0)$ ✓ | Mid (Gaussian errors only) | High ✓ | Mid | Mid | Mid |
| In situ NAT | | Loss | High ✓ | Mid (Gaussian errors only) | $O(TO)$ ✓ | Mid (Gaussian errors only) | High ✓ | High ✓ | Mid | Mid |
| In silico SAT (Ours) | | Loss & Sharpness ✓ | High ✓ | High ✓ | $O(T_0)$ ✓ | High ✓ | High ✓ | High ✓ | High ✓ | High ✓ |
| In situ SAT (Ours) | | Loss & Sharpness ✓ | High ✓ | High ✓ | $O(TO)$ ✓ | High ✓ | High ✓ | High ✓ | High ✓ | High ✓ |

$T_0$ is the time to convergence for backpropagation[58].
Generality means whether the method is general to different physical systems.
Scalability means whether the method is still effective when the physical system and task complexity scale up.
PAT physical-aware training, DAT dual-adaptive training, GA genetic algorithm, SO surrogate optimization, ES evolutionary strategy, NAT noise-aware training, SAT sharpness-aware training.

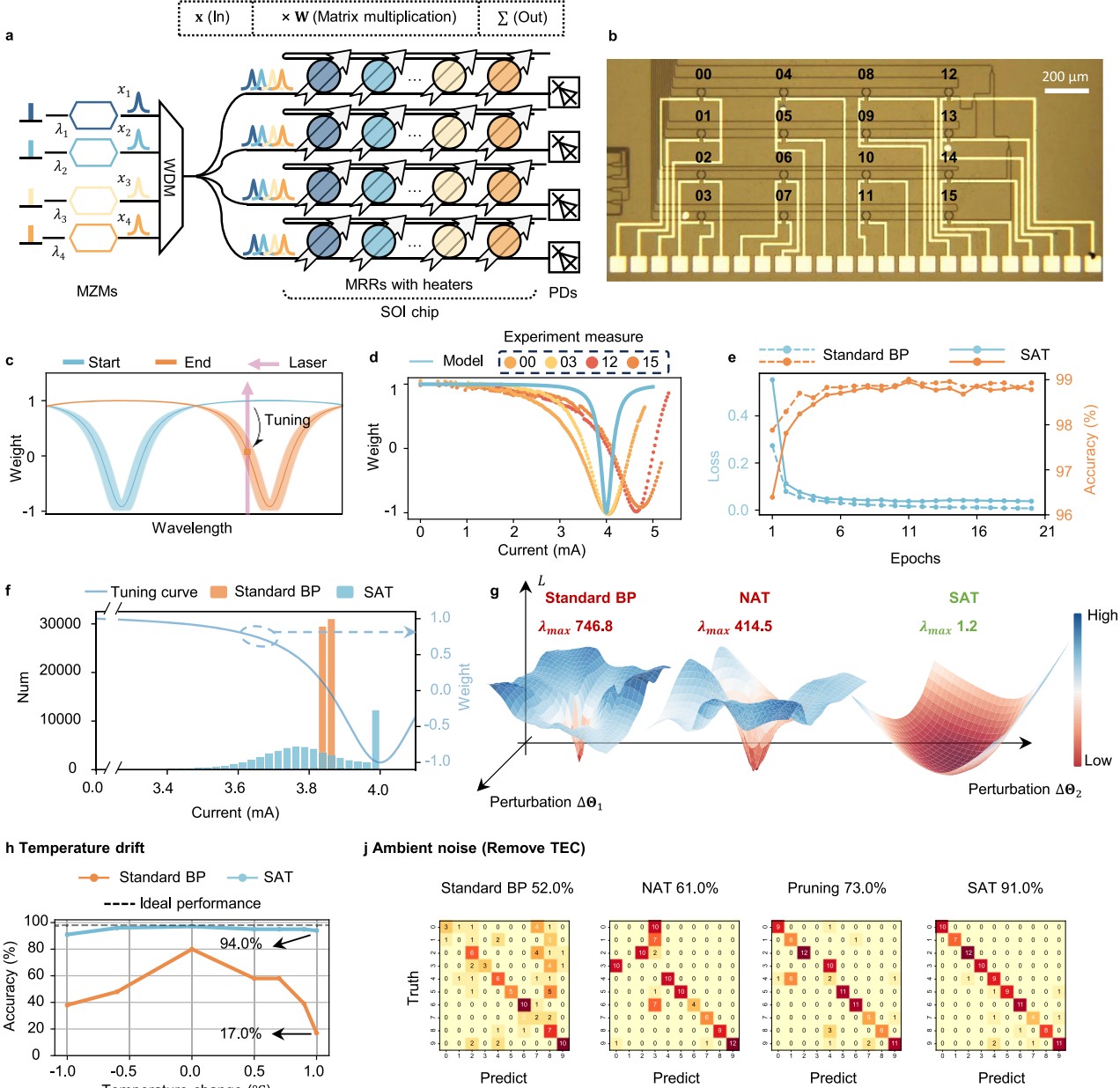

**Fig. 2 | In silico training results in MRR-based PNNs. a** Implement matrix multiplication on the MRR weight bank. **b** Picture of the manufactured MRR weight bank. **c** Single MRR spectrum changes through thermal tuning. **d** Ideal MRR tuning curve model and experimentally measured MRR tuning curves. **e** Loss and classification accuracy change versus training epochs. **f** Current distribution contrast histogram. **g** Trained model loss landscapes. **h** Experimental inference accuracy with temperature change from 21 to 23 °C. **j** Experimental measurements of inference accuracy without using TEC. MZM Mach-Zehnder modulator, WDM wavelength division multiplexer, MRR microring resonator, SOI silicon on insulator, PD photodetector, BP backpropagation, SAT sharpness-aware training, TEC thermoelectric control.

the stability of individual components but also enhances the overall system's stability. Consequently, SAT achieves the highest accuracy at 91.0%.

Additionally, we extend the application of our approach to much more challenging tasks, including image classification, compression, reconstruction, and generation, as shown in Fig. 3a–m. Our results show that PNNs trained with SAT maintain high accuracy in all these tasks under temperature variations within ±0.25 °C, a thermal fluctuation level in current co-packaged optics[50]. In contrast, PNNs trained using standard methods suffer performance degradation and fail under the same conditions. In the image classification task, we apply the ResNet18 model in the MRR-based PNN to classify the CIFAR-10 dataset. The model's robustness is evaluated by varying the operating

temperature. As shown in Fig. 3b–d, SAT maintains stable accuracy across ±0.25 °C temperature variations, whereas standard BP fails. Specifically, under a −0.25 °C shift, BP's accuracy drops to 10.6%, while SAT maintains 83.4%, nearly matching its performance under nominal conditions. Image compression and reconstruction tasks use an autoencoder framework. We first compress CIFAR-10 images into a latent space with an MRR-based encoder, followed by digital decoding. As shown in Fig. 3f–h, for the SAT-trained model, even under ±0.25 °C shifts, the reconstruction quality, assessed using mean squared error (MSE), remains at around 0.023, comparable to its nominal performance (0.013). In contrast, the BP-trained model fails. Under a −0.25 °C shift, the reconstruction collapses with an MSE of 0.529. Next, we implement a generative adversarial network where the MRR-based

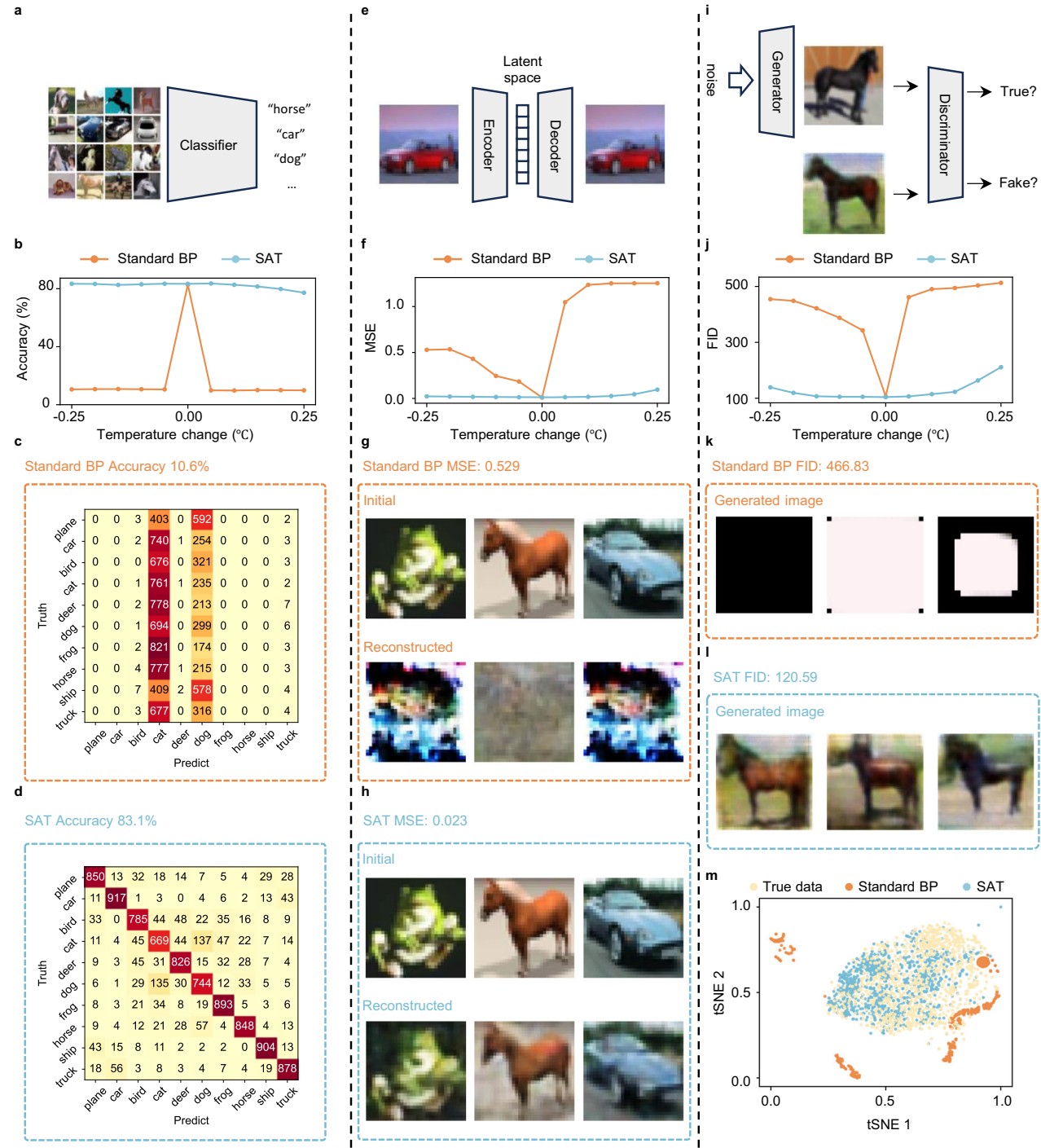

**Fig. 3 | Results of CIFAR-10 image classification, compression, reconstruction, and generation using MRR-based PNNs. a** Schematic diagram of CIFAR-10 image classification. **b** Neural network inference accuracy with a 0.5 °C temperature variation. **c** Confusion matrix for CIFAR-10 classification with standard BP under a −0.25 °C temperature shift. **d** Confusion matrix for CIFAR-10 classification with SAT under a −0.25 °C temperature shift. **e** Schematic diagram of CIFAR-10 image compression and reconstruction. **f** MSE of CIFAR-10 image reconstruction under a 0.5 °C temperature variation. **g** CIFAR-10 image reconstruction with BP under a −0.25 °C temperature shift. **h** CIFAR-10 image reconstruction with SAT under a −0.25 °C temperature shift. **i** Schematic diagram of CIFAR-10 image generation using a generative adversarial network. **j** FID of generated images under a 0.5 °C temperature variation. **k** CIFAR-10 image generation performance with BP under a −0.25 °C temperature shift. **l** CIFAR-10 image generation performance with SAT under a −0.25 °C temperature shift. **m** t-SNE visualization of generated image distributions under a −0.25 °C temperature shift. BP backpropagation, SAT sharpness-aware training, MSE mean squared error, FID Fréchet Inception Distance, t-SNE t-distributed stochastic neighbor embedding.

PNN serves as the generator. The quality of the generated images is evaluated using Fréchet Inception Distance (FID)[46]. As shown in Fig. 3j, SAT ensures stable FID values under temperature shifts, whereas BP suffers from severe degradation. Figure 3k, l further highlights the robustness of SAT. The images generated by SAT remain clear under a −0.25 °C shift, while BP-generated images degrade. Additionally, the t-SNE visualization (Fig. 3m) further confirms that SAT-generated images maintain distributional consistency with the original dataset,

unlike those generated by BP. Details are clarified in Supplementary Note 4.4.

## In situ SAT: facilitating transferable in situ training

This section addresses the transferable learning challenge of in situ training: the trained results are specific to a particular system and cannot be easily transferred to other systems, even those fabricated with the same design, due to fabrication variances. This lack of transferability is a drawback for modern AI hardware, where training is typically performed on a single high-end computer or cloud-based cluster, and the trained parameters are then deployed across multiple edge devices for inference. Here, we demonstrate that SAT can be combined with in situ training approaches to overcome this major drawback and enable transferable in situ training.

We demonstrate SAT on the MZI meshes based NN through numerical simulations using the library provided by Ziyang et al.[28]. In MZI mesh-based NNs, the basic building block is a $2 \times 2$ MZI, which consists of two 50-50 beam splitters and two phase shifters. This block performs an arbitrary $2 \times 2$ unitary transformation by configuring two phase shifters. By cascading multiple MZI blocks, the MZI mesh can achieve arbitrary unitary transformations[51]. Programming such a system remains challenging, as the model is difficult to characterize due to unknown initial phases caused by phase errors in the phase shifters. Consequently, many in situ training approaches have been proposed to program such systems. However, wafer-level manufacturing errors make the trained results specific to only the trained system but not transferable to other systems. For example, as pointed out by ref. 52, variations of the beam splitter as small as 2%, which is a typical wafer-level variance, can degrade the NN accuracy by nearly 50%. We demonstrate in situ SAT on a simulated digital-optical hybrid NN, following the setup in ref. 28. The digital NN consists of two convolutional layers followed by two fully connected layers, with the functions of feature extraction and image down-sampling. The optical NN is a $64 \times 64$ MZI mesh followed by a square detection layer, serving as the output layer. The first 10 output ports are used to generate classification output on the MNIST dataset.

We evaluate six approaches under fabrication variances: (1) in silico training with standard BP on a digital model without considering fabrication variances; (2) in silico NAT, which injects synthetic noise during training; (3) physical-aware training (PAT), an in situ approach that leverages the actual PNN system for forward propagation and a proxy model for gradient estimation; (4) dual-adaptive training (DAT), an in situ approach modified from PAT, capable of deriving a more accurate proxy model for improved gradient estimation; (5) in silico SAT, trained in silico on a digital model and deliberately ignoring fabrication variances; (6) an in situ SAT, which adopts the PAT framework but adding searching for the sharpness minimum. Detailed setups for these approaches are clarified in Supplementary Note 5. The purpose is to verify: (1) In silico SAT can address model-reality gap and even outperforms in situ training methods; (2) SAT can integrate with in situ training, particularly PAT, and improve their performance in light of inaccurate modeling in PAT; (3) In situ SAT can enable transferable learning to other devices, overcoming the major drawback of current in situ training methods.

To simulate the fabrication errors, we generate phase error $\sigma_{ps}$ and splitting error $\sigma_{bs}$ sampling from a Gaussian distribution with a mean of 0 and variance of 0.15, and add these errors to the MZI mesh model. As shown in Fig. 4c, when using in silico training with standard BP, these fabrication errors cause an accuracy reduction from 97.4 to 68.6%, corresponding to an error rate of 31.4%. This is because the fabrication variances create a substantial mismatch between the ideal model used for training and the actual system. The training time is 356.1 s. Next, we apply the NAT training method. During training with NAT, the noise we add to the phase parameter follows a Gaussian distribution with a mean value of 0 and a standard deviation of

0.01 rad. The values are chosen to maximize robustness without decreasing the accuracy. Additionally, we add Gaussian noise to intentionally increase the trained model's robustness, and the trained accuracy is only 79.8%, corresponding to an error rate of 20.2% caused by noise injection. The training time is 338.2 s.

To address the model mismatch, we turn to in situ training methods, including PAT and DAT, which loop the device in the training process. In PAT, the forward pass uses the system with fabrication errors, while the backward model is an ideal MZI mesh without error. Despite using in situ training, under large fabrication errors, PAT achieves an accuracy of only 70.2% with a training time of 393.8 s. The low accuracy is due to the approximated gradients from the ideal model deviating from the true gradients, which hinders the loss minimization. To obtain a more accurate proxy model for BP, DAT combines the ideal model with an additional digital model using the data-driven method. DAT brings the approximated gradients closer to the real gradients, therefore improving the accuracy to 94.6%. But this improvement is at the cost of adding six additional systematic error prediction networks (SEPNs), with the training time increasing to 1272.7 s. The results are consistent with ref. 28.

We also find the model trained by DAT is highly sensitive to noise and perturbations. We use the maximum eigenvalue of the Hessian matrix $\lambda_{max}$ to evaluate the trained model, as a large $\lambda_{max}$ indicates high sensitivity to parameter changes. The $\lambda_{max}$ for DAT is 252.4, nearly three times that of the model trained in silico with standard BP, which has a $\lambda_{max}$ of 93.5. This result implies that while DAT provides accurate gradients and improves classification performance, this comes at the cost of reduced robustness. As a result, the trained system will be highly vulnerable to perturbations and noise after deployment, and the trained parameters cannot be transferred to other systems, even those fabricated with the same design.

Next, we evaluate in silico SAT and in situ SAT (which is adapted from the PAT framework but with additional training to search for the sharpness minimum). We first implement SAT trained in silico on an ideal model. The training time is 513.8 s, which is longer than that of in silico training with standard BP due to the need for two steps of BP. However, the trained system is much more robust to the fabrication errors, achieving an accuracy of 92.3% even under large fabrication errors. Remarkably, this performance not only outperforms in silico training methods but also outperforms in situ training methods, including PAT and DAT. Moreover, the Hessian max eigenvalue $\lambda_{max}$ is only 0.3, which is lower than other methods under comparison. The result shows the trained model has high robustness compared to others.

To further improve accuracy, we perform in situ SAT by combining SAT with PAT, resulting in an accuracy of 97.1%. This outcome suggests that incorporating SAT effectively addresses the modeling inaccuracy in PAT: in situ SAT demonstrates the ability to achieve high training performance even with inaccurate approximating gradients. Moreover, in situ SAT also exhibits high robustness. The Hessian max eigenvalue $\lambda_{max}$ is only 0.6, indicating that SAT improves the robustness of the trained system.

This robustness allows the parameters obtained through in situ training to be transferred to other devices without a loss in accuracy. To evaluate this, we apply the trained model to other devices with different randomly generated phase errors and splitting ratio errors. We systematically increase the errors in both phases and the splitting ratio and assess classification accuracy, with results shown in Fig. 4d. SAT is capable of maintaining high accuracy at 95.2%, indicating that SAT can facilitate transferable in situ training. In contrast, the system trained by DAT decreases to 58.6% when the phase and splitting ratio errors reach 0.15.

In summary, SAT enhances both in silico and in situ training schemes. When trained in silico, SAT achieves high accuracy under large fabrication errors, surpassing state-of-the-art in situ methods. Moreover, integrating SAT with in situ training (such as PAT) further

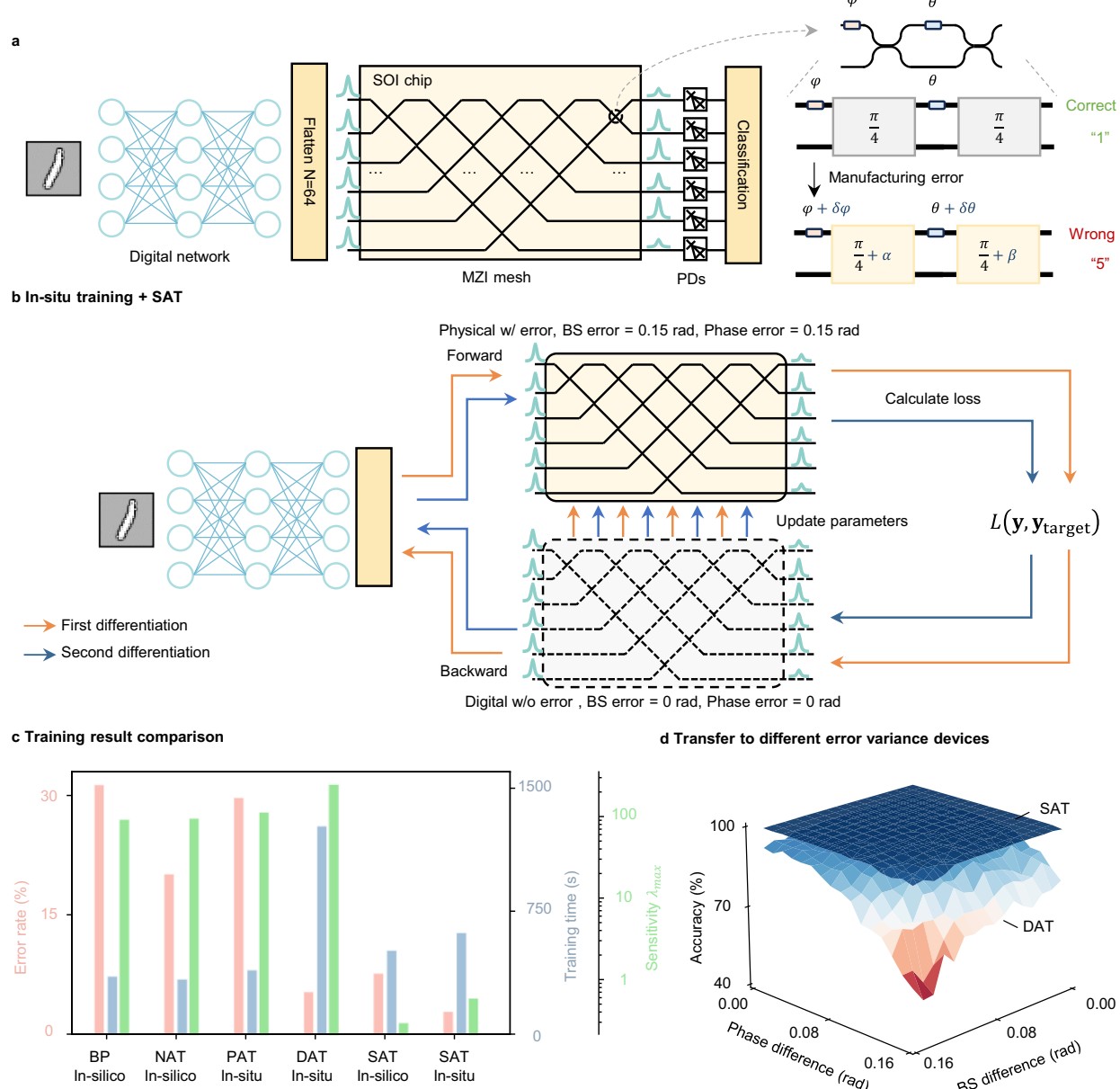

**Fig. 4 | In situ training results in MZI-based PNNs. a** Schematic diagram of a digital-optical hybrid network and an illustration of manufacturing error. **b** Schematic diagram for implementing SAT in in situ training. **c** Training result performance comparison. **d** Inference performance comparison when transferring trained parameters to different error variance devices. MZI Mach-Zehnder interferometer, PNN Photonic neural network, SOI Silicon-on-insulator, PD Photodetector, SAT sharpness-aware training, BP backpropagation, NAT noise-aware training, PAT physical-aware training, DAT dual-adaptive training, BS beam splitter.

improves accuracy. Importantly, SAT enables the trained parameters to be transferable to other devices, addressing a key limitation of current in situ training methods.

**Extending SAT to PNNs without explicitly known models**
We further demonstrate our method's effectiveness in diffractive optical NNs[53]. The training is performed in silico. Unlike the MRR-based PNN system or MZI-based PNN system, where the relationship between the weights and the control parameters is known (though not entirely precise), in diffractive optical NNs, the relationship between the misalignment factors, such as rotation angle and the controlling parameters, is not explicitly known.

In diffractive optical NNs, as shown in Fig. 5a, b, an NN layer is decomposed into multiple matrix multiplications and realized by an

imaging system. In the experiment setup depicted in Fig. 5c, element-wise multiplication is achieved by first encoding the image to be processed on an organic light-emitting diode (OLED) serving as the light source. The optical image then passes through a spatial light modulator (SLM), which performs pixel-wise intensity modulation on the input image, thus realizing element-wise multiplication[53]. (Details are explained in the Supplementary Note 6.1). In this setup, misalignment between the OLED and SLM can severely degrade system performance. Our experiment shows that 1° rotation angle misalignment between the OLED and the SLM can cause a drop in accuracy, from 98.0 to 43.0% on the MNIST dataset. Other misalignments include $x$- and $y$-axis shifts between the OLED and the SLM, and the distance change between them (i.e., $z$-axis shift) that results in image scaling.

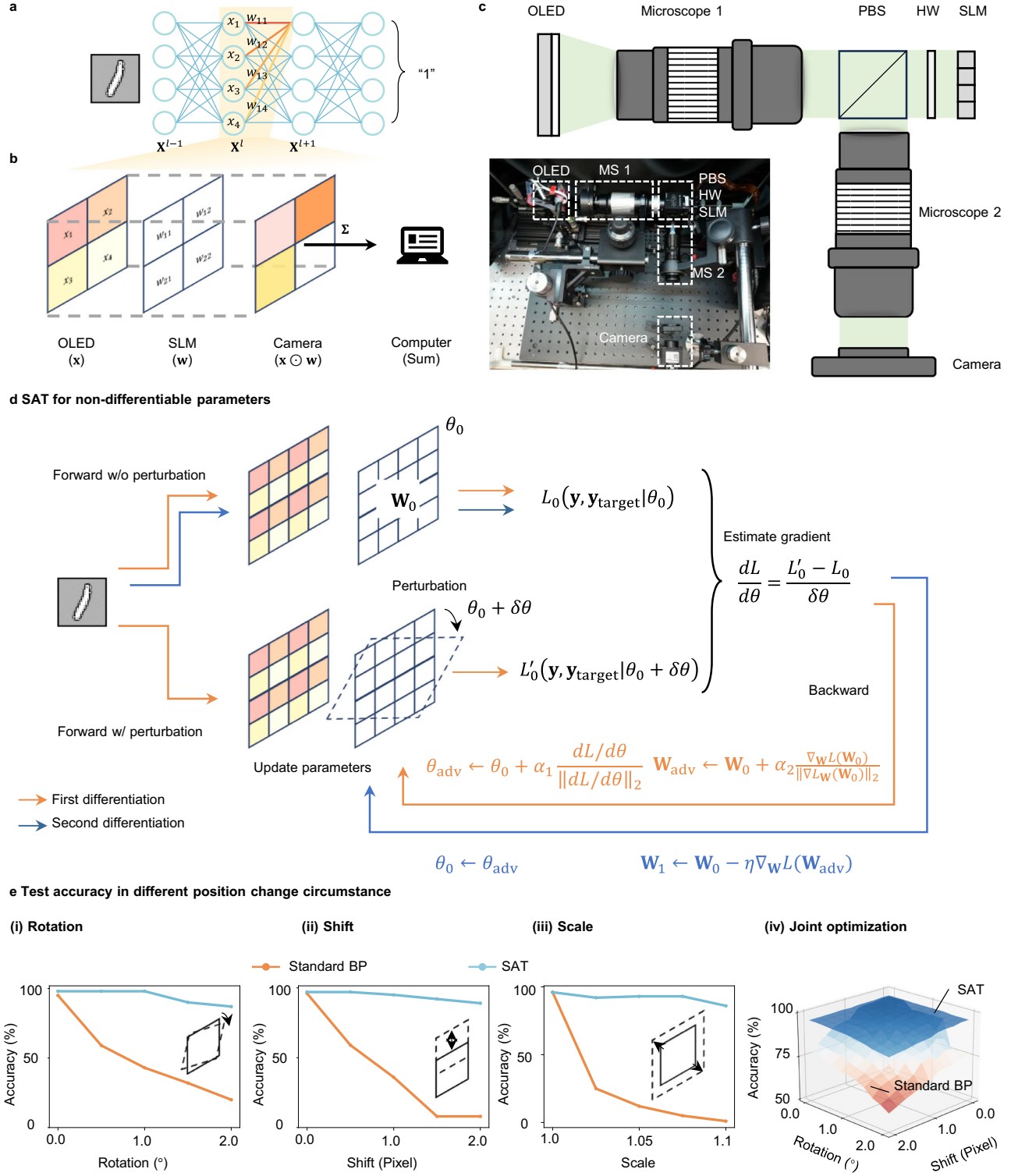

**Fig. 5 | Experimental results in diffractive-optics-based NNs. a** Decompose neural network inference into multiple matrix multiplications. **b** Deploy matrix multiplication in free-space optical computing systems. **c** Schematic diagram and picture of the experimental setup. **d** Detailed SAT optimization process for non-differentiable parameters. **e-i** Experimental inference accuracy with rotation angle change from 0.0° to 2.0°. **e-ii** Experimental inference accuracy with shift pixel number change from 0.0 to 2.0. **e-iii** Experimental inference accuracy with scaling number change from 1.0 to 1.1. OLED organic light-emitting diode, SLM spatial light modulator, MS microscope, PBS polarization beam splitter, HW half-wave plate, SAT sharpness-aware training, BP backpropagation.

In such a system, there is no explicit model between the mis-alignment factors and controlling parameters, i.e., characterizing the model y = $F(x; \mathbf{\Theta})$ is difficult, where $\mathbf{\Theta}$ represents the misalignment factors such as rotation angle and axis shifts between the OLED and the SLM. To address this, we propose a modified SAT method in which the gradient of the loss with respect to the misalignment factor is estimated by the finite difference method. The optimization flow is shown in Fig. 5d using rotation angle $\theta$ as an example. During the first

automatic differentiation process, as noted by the yellow arrows, to estimate the gradient, a small perturbation $\delta\theta$ is added to the rotation angle, and the loss is calculated twice, once without this small perturbation and once with it. The gradient with respect to the rotation angle is then estimated using the finite difference method, as depicted in Fig. 5d. After obtaining this gradient $dL/d\theta$, the rotation angle and weights are updated according to Eq. (4),

$$
\begin{aligned}
\theta_{\text{adv}} &= \theta_0 + \alpha_1 \frac{dL/d\theta}{\|dL/d\theta\|_2} \\
\mathbf{W}_{\text{adv}} &= \mathbf{W}_0 + \alpha_2 \frac{\nabla_{\mathbf{W}} L(\mathbf{W}_0)}{\|\nabla_{\mathbf{W}} L(\mathbf{W}_0)\|_2}
\end{aligned}
\tag{4}
$$

Here, $\theta_0$ and $\mathbf{W}_0$ denote the initialized rotation angle and weights, $\theta_{\text{adv}}$ and $\mathbf{W}_{\text{adv}}$ represent the adversarial rotation angle and weights that reflect the maximum loss within the initial parameters' neighborhood. The hyperparameters $\alpha_1$ and $\alpha_2$ determine the perturbations added to the rotation angle and weights, respectively. The functions of $\alpha_1$ and $\alpha_2$ are equivalent to $\alpha$ in Eq. (3). Because parameters $\theta$ and weights $\mathbf{W}$ are optimized separately, we use different subscripts to indicate the difference.

Next, we maintain the rotation angle at $\theta_{\text{adv}}$ during the second automatic differentiation process and conduct forward propagation, as noted by the blue arrows. Then the gradient with respect to weights is calculated again at $\mathbf{W}_{\text{adv}}$. This calculated gradient is used to update the initial weights $\mathbf{W}_0$. Finally, the rotation angle is reset to the initial value $\theta_0$ as the rotation angle remains constant once calibrated in the real-world system. The training is performed on a computer in an in silico version without considering side effects like dark noises. (The detailed training process is illustrated in Supplementary Note 6.2).

We apply the same optimization approach to address the issues of shift and scaling. The training is performed in silico. After training, we deploy the trained model in our experimental setup to perform NN inference. During the SAT training, we assume perfect alignment between the OLED and SLM. This simplification is intentional, as SAT-trained systems are expected to be inherently robust to implementation errors, even when those errors are unknown. As a result, we expect the system trained by SAT to maintain high accuracy in the presence of practical misalignments such as rotation, shift, or scaling.

To quantitatively assess the tolerance of our method to such misalignment errors at the implementation stage, we follow the approach in ref. 28 and apply affine transformations to introduce controllable deviations (rotation, shift, and scale). We then evaluate the system's accuracy under these conditions. This procedure enables us to systematically assess the resilience of the trained model to realistic imperfections in the physical setup. The experimental results shown in Fig. 5e illustrate that SAT improves the model's robustness against the misalignment issues in the free-space computing system. Specifically, SAT maintains MNIST classification accuracy at 98.0% at 1° rotation angle misalignment without accuracy reduction, compared to 43.0% using the standard training method. In terms of pixel-wise shift, we intentionally shift the pixels of the SLM from 0 to 2 pixels in the y-axis direction. The results show that SAT achieves 97.0% accuracy at 1-pixel shift, while the standard training method attains only 36.0% accuracy. Finally, we scale the image size in the SLM to emulate the z-axis shift, with the scaling factor sweeping from 1.0 to 1.1. Our experimental results show that the standard training method achieves only 12.0% accuracy, but SAT improves the accuracy to 93.0% with the scaling factor at the level of 1.05. These results demonstrate that SAT is generally applicable to different computing systems, even for those systems without having an explicitly known model. The modified SAT can successfully address this issue by using the finite difference method to estimate the gradient of the loss with respect to those parameters.

## Discussion

In summary, we propose a training method that addresses key limitations in both in silico and in situ training of PNNs.

Through demonstrations on three typical PNNs, an MRR-based PNN, an MZI mesh-based PNN, and a diffractive optical NN, we highlight several distinguishing features of SAT that offer clear advantages over current in silico and in situ training methods: (1) SAT enables accurate in silico training of a PNN, even with an imprecise digital model and under deployment errors; (2) SAT allows systems trained in silico to outperform those trained in situ, even in the presence of fabrication errors; (3) systems trained using SAT exhibit high robustness to environmental fluctuations and noise; (4) SAT can be integrated with in situ training methods, facilitating transferable in situ training across different devices; and (5) SAT is universally applicable to real-world challenging tasks and various PNNs, regardless of whether their models are explicitly known or unknown. Overall, SAT offers a practical, effective, and computationally efficient solution for training and deploying PNNs in real-world applications.

### Training cost

While SAT has a slight cost in additional automatic differentiation during training, it remains far more efficient than methods requiring extensive experimental measurements or the training of a large digital twin model. However, the additional training steps can be further reduced using the approach proposed in ref. 54, which reduces the sharpness with almost zero additional computational cost by calculating the Kullback–Leibler divergence between the NN's outputs using current weights and past weights, as a substitute for SAT's sharpness measure. This approach will offer the benefits of reduced sharpness without extra overhead.

### Joint parameter optimization

During the training of diffractive-optics-based NNs, we perform optimization for each parameter (rotation angle, shift, and scale factor) and evaluate its impact on inference performance. We can also jointly optimize these parameters. Here, we extend the verification of our method to jointly optimizing multiple control parameters. The simulation details and results are depicted in the Supplementary Note 6.3. We propose a sequential joint optimization strategy: optimizing the rotation angle in epoch 1, the pixel shift in epoch 2, and the scale factor in epoch 3. This approach enables joint consideration of all parameters while maintaining the same computational complexity as optimizing a single parameter at a time. Our simulation results show that joint optimization can simultaneously enhance the model's robustness over different misalignments, while the robustness trained by joint parameter optimization is comparable to that of single parameter optimization.

### Compatible with nonlinear systems

Although our experimental demonstration uses a linear system and assumes digital nonlinearity in simulations, SAT can naturally be extended to systems with physical nonlinearity. Because the gradient approximation process is the same for linear and nonlinear systems. To prove this, we also include an additional simulation study on a nonlinear deep diffractive neural network described in ref. 21. The simulation details and results are included in Supplementary Note 6.4. The simulated system consists of two programmable linear layers and two nonlinear layers. The first nonlinear layer is implemented using a saturable absorber, which introduces an intensity-dependent transmission while preserving the optical phase. And the second nonlinear component is the PD array, which introduces a quadratic nonlinearity during the optical-to-electrical conversion process. We train the physical network using both standard BP and our SAT. During training, we employ the finite difference method to approximate the loss gradient with respect to the rotation angle (which acts as a trainable control

parameter), and we intentionally introduce perturbations in rotation to evaluate robustness. The simulation results show our method maintains high classification accuracy under rotation angle variations.

## Methods

### Dataset

The MNIST dataset contains 70,000 grayscale images of handwritten digits (0–9)[48]. These include 60,000 training examples and 10,000 examples, each formatted as a $28 \times 28$ pixel grid with corresponding digit labels. The CIFAR-10 dataset contains 60,000 color images categorized into 10 distinct classes (airplane, automobile, bird, cat, deer, dog, frog, horse, ship, truck), including 50,000 training examples and 10,000 test examples[55]. Each image is formatted as a $32 \times 32$ pixel grid (RGB color channels) with corresponding class labels.

### Device fabrication

The device is fabricated in the commercial multi-project wafer process from Applied Nanotools. The silicon thickness of this process is 220 nm, and the buffer oxide layer thickness is 2 µm. The MRR weight bank contains 16 MRRs and is arranged in a $4 \times 4$ matrix. Each row sub-weight bank consists of four MRRs coupled with two bus waveguides in an add/drop configuration. The radii for each MRR are around 20 µm, and we intentionally introduce a slight 10 nm radius difference to avoid resonance collision. And the result shows that resonance wavelengths are roughly spaced by 0.8 nm. The gap between the MRR and the bus waveguide is 100 nm. MRR Q factor is around $10^4$. Circular metal heaters (TiW) are built on top of each MRR for thermal weight tuning. Metal (Al) vias and traces are deposited to connect the heater contacts of the MRR weight bank to the electrical metal pads. The heater resistance is around 375 Ω. The tuning efficiency is around 0.53 nm/mW.

### MRR-based PNN experimental setup

During the experiment, we selected four MRRs with the labels "00," "03," "12," and "15." Before the experiment, we first actuate initialization currents to align the resonance wavelengths of "00" & "03" and "12" & "15." After initialization, the resonance wavelengths of "00" & "03" are both 1542.7 nm, and the resonance wavelengths of "12" & "15" are both 1544.8 nm. A TEC module is set to stabilize or intentionally change the temperature. The laser (KG-TLS-C-13-50-P-FA, Conquer) wavelengths are set to 1543.0 and 1545.0 nm, respectively. Different wavelength lights are independently modulated using electrical-optical modulation (KG-DDMZM-RF-B, Conquer). The modulation signals are set with a baud rate of 20 MBaud (AFG 31000 SERIES, Tektronix). The different wavelength lights are then combined through a 50:50 coupler. Before coupling into the chip, the combined light is amplified through an EDFA (FA-30, PriTel) to around 12 dBm. The coupling loss of the chip is around 14 dBm. The two-channel output lights are then detected by the PD (PDT0313-FC-A, HP) at the through port. During the matrix multiplication, the negative weights are mapped to the tuning range [0,1].

### Details about training the digital-optical hybrid network

The digital-optical hybrid network is built based on the open-source library "DAT MPNN"[28] and "Neurophox"[56]. The network is a modified version of the classic LeNet-5 architecture, designed for image classification tasks, with an optical MZI mesh layer for classification. Details are clarified in Supplementary Note 5.

Specifically, we test 6 different training methods: in silico standard BP, in silico NAT, in situ PAT, in situ DAT, in silico SAT, and in situ SAT. The training device is an Nvidia 3070Ti, and the training time is measured by directly calculating the difference between the start time and end time of the program run. The in silico standard BP involves training the ideal model directly for 40 epochs. During training with NAT, the noise we add to the phase parameter follows a Gaussian distribution with a mean value of 0 and a standard deviation of 0.01 rad. The aim is to maximally increase the robustness without decreasing the accuracy. For PAT, training involves performing inference on the physical system and BP on the ideal system. The physical system deviates from the ideal system due to static phase errors and splitting errors. The total number of training epochs is 40 for accuracy and loss to converge.

In the case of DAT, an additional digital network, SEPN, is introduced to compensate for the gap between the physical system and the ideal system. DAT requires more training epochs to converge, totaling 60 epochs. During the first 30 epochs, the training process is the same as the PAT, and SEPN is trained to reduce the output difference between the physical system and the ideal system, but it is not used for gradient calculations. In the subsequent 30 training epochs, SEPN training stops, and SEPN assists the ideal model in providing accurate gradients. To minimize the gap as much as possible, we use 6 complex mini-UNet networks connected in parallel to extract multiscale features. The mini-UNet first downsamples the input image from its original size ($H$, $W$) to ($H/2$, $W/2$). It then further reduces this to ($H/4$, $W/4$). Each of these three feature maps, at different spatial resolutions, is processed independently by a dedicated complex convolutional NN. Finally, the processed feature maps are fused and passed through a series of transposed convolutions to reconstruct an output image with the same size as the original input. The detailed network of mini-UNet is the same as that in ref. 28 and is illustrated in Supplementary Note 5 and Fig. S17.

For in silico SAT, standard BP is still used for differentiation, but differentiation is performed twice per epoch. The total number of training epochs remains 40. For in situ SAT, differentiation is provided by the ideal model, which is the same as PAT, and differentiation is performed twice per epoch. The total number of training epochs is 40. The initial learning rate for training the PNN is 0.1, which is decayed by 0.5 at half of the maximum training epoch number. The learning rate for training the SEPNs is set at 0.001 for constant. The cross-entropy loss function is selected to calculate the task loss, and the MSE loss function is selected for adaptive training of the SEPNs.

### Free-space experimental setup

In the free-space experimental setup, a custom OLED display (Sony ECX335S) is used, encoding pictures as the light source. Only the green channel is utilized in the experiments. The display has a resolution of $1920 \times 1080$ and a refresh rate of 60Hz. We develop custom Python control codes to load images. The display features 256 distinguishable brightness levels, corresponding to an 8-bit resolution. We achieve the weights through light intensity modulation using an SLM (LCOS-SLM, HDSLM80R Plus, UPO labs) with a polarizing beam splitter (PBS) and a half-wave plate (HW) in a double-pass configuration. A zoom-lens microscope (BYH0330, Inseinlifung) is used between the OLED and SLM to match the pixel sizes between the OLED (5 µm) and SLM (8 µm). The light is then projected onto a camera using a second microscope (SHL-0745C, Shhunhuali). A CMOS camera (MV-SUF89OGC/M, Mindvision) captures the modulated light field as an image. The image is separated into different regions, and the sum of the intensities from these regions represents the matrix multiplication result.

## Data availability

The raw datasets are all publicly available. MNIST[48] and CIFAR-10[55] datasets are downloaded from previous works. Source data are provided via Zenodo at https://doi.org/10.5281/zenodo.17905440[57].

## Code availability

The source code supporting the findings of this study has been deposited in the GitHub repository at https://github.com/cuhkhuangslab/Sharpness-Aware-Training/tree/main. The repository is publicly accessible under the MIT License, permitting reuse and

modification with appropriate citation. The software code is also available via Zenodo at https://doi.org/10.5281/zenodo.17905440[57].

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

## Acknowledgements

This work was supported by NSFC 62405258, RGC ECS 24203724, GRF 14208925, YCRG C4004-24Y, YCRG C1002-22Y, STG3/E-404/24-N, ITF ITS/237/22, National Key Research and Development Program of China 2024YFE0203600, NSFC/RGC N_CUHK444/22.

## Author contributions

Z.L., T.X., and C.H. conceived the ideas. T.X. and S.L. built the MRR-based PNN computing system with the help of Q.X. and B.W. Z.L. and L.F. built the diffractive-optics-based NN computing system. T.X., S.L., Z.L., and C.H. designed and conducted the experiments of the MRR-based PNN. Z.L., L.F., T.X., and C.H. designed and conducted the experiments of the diffractive-optics-based NN. T.X., Z.L., S.L., L.F., Q.X., and B.W. analyzed the results. D.W. designed the PCB board and performed the chip packaging. T.X., S.L., and C.H. wrote the manuscript with input from all authors. C.H. supervised the research and contributed to the general concept and interpretation of the results. All the authors discussed the data and contributed to the manuscript.

## Competing interests

The authors declare no competing interests.
