## [Transparent Peer Review file · Nature Communications]

Physical Neural Networks using Sharpness-Aware Training

Corresponding Author: Professor Chaoran Huang

Version 0:

Reviewer comments:

Reviewer #1

(Remarks to the Author)

Xu et al. propose to apply a technique from conventional machine learning, <<sharpness-aware training>> (SAT), to training physical neural networks (PNNs), and empirically observe that it works well when applied to some optical neural networks. While SAT was initially introduced to improve generalization, here the authors find that it also makes training more robust to noise/drifts in physical systems.

I believe the idea is sound and the results show that SAT could be generally useful in PNNs. Because of the broad utility of the approach, I strongly recommend publication of the manuscript in Nature Communications subject to the authors addressing the following comments.

1. The paper seemingly overclaims novelty: SAT was not invented by the authors, but rather was previously reported in references [42] and [50]. The current abstract says: "...through a novel technique called Sharpness-Aware Training". But the technique is not novel! (Indeed it was reported 4 years ago, in the most prominent machine-learning conference.) It would be more appropriate for the abstract (and other parts of the paper) to describe what they have done as apply the (established; not novel) technique SAT to the training of PNNs and observe that it results in improved robustness to model imperfections and PNN noise/drifts etc.
2. The standard approach to training a physical (analog) neural network for which one has a fairly accurate model is to perform training in a digital computer but to inject noise during training (sometimes called noise-aware training: doi:10.1109/TED.2021.3089987 ; review: doi:10.1002/aisy.202200029). For Figure 2j and for Figure 5c, it would be useful if the authors could add a comparison to noise-aware training (NAT) or related standard noise-injection techniques. I also think it would be useful for physics-aware training (PAT) to be added as a point of comparison in Figure 2j.
3. The authors have not provided the code to replicate their results. Especially for a methodology paper, this is unacceptable and against Springer Nature's stated policy on code availability. The authors should provide the code in a GitHub (or similar) repository. At the very least, this repository should contain everything needed to rerun the simulations reported in Figure 5.

Reviewer #2

(Remarks to the Author)

The article introduces a novel training approach, Sharpness-Aware Training (SAT), aimed at enhancing both the training process and deployment robustness of physical neural networks (PNNs). I appreciate the insight that the authors establish a connection between sharp minima in the loss landscape and the poor robustness observed in physical systems. I believe this work has the potential to become a foundational methodology for training PNNs—specifically, by incorporating a sensitivity term of the objective function into the loss formulation to promote the learning of more robust parameters. In addition, I have a few questions and suggestions that I hope the authors can clarify, which I believe would further improve the quality of the paper.

1. When the influence of certain parameters on the physical process is not explicitly known, approximating the gradient via finite differences offers an effective way to bypass the need for explicit modeling. Figure 3(e) illustrates the separate optimization processes for rotation, shift, and scale parameters. Have the authors considered jointly optimizing these parameters instead of optimizing them independently?

2.Lines 390–391 state: “During the experiment, we use the affine function to quantitatively adjust rotation, shift, and scaling parameters.” Is it truly feasible to achieve such precise control over angle and pixel-level shifts in a physical experimental setup? Figure 3e appears to show results from a real experiment, but in practice, these deviations—especially angular or spatial misalignments—are typically not precisely known or controlled.

3.Line 297 states that “the system trained with standard BP degrades significantly from 80.0% to 7.0% as the temperature varies from 21°C to 23°C.” However, I could not clearly observe this drastic performance drop in Figure 2(h). Could the authors clarify this discrepancy or improve the clarity of the visualization?

4.I am curious about the tolerance limits of the proposed method. Specifically, how large can the deviations (e.g., phase errors, temperature shifts) be before the performance of the SAT-trained model degrades significantly or fails altogether?

5.The method effectively mitigates performance degradation caused by various noise sources, but how is the trade-off between sharpness and loss handled in practice? How should the hyperparameter α be chosen? A similar issue arises with the self-defined hyperparameters α_1 , r , μ , and ρ . It would be beneficial if the authors could provide a principled guideline or sensitivity analysis.

6.Minor issues: In Figure 2(h), it would be more accurate to describe the performance shown as the ideal performance rather than the theoretical limit. In Equation (4) of the main text, the numerator appears to be missing the letter “L”.

7.I strongly encourage the authors to consider releasing the code, even if only for the MZI-based implementation, which would significantly enhance the impact and reproducibility of this work.

Reviewer #3

(Remarks to the Author)

This manuscript introduces the use of Sharpness-Aware Minimization (SAM), a technique originally developed in the machine learning domain, to improve the robustness of physical neural networks. By simultaneously minimizing the loss and its sharpness, the authors aim to eliminate the need for in-situ training, thereby enabling the network to tolerate environmental fluctuations and fabrication variance. The effectiveness of this approach is experimentally evaluated across three distinct hardware platforms.

While the manuscript presents a technically sound implementation and provides extensive experimental validation, I have the following major concerns regarding its novelty and overall impact:

1. Lack of originality in methodology: The training is entirely performed in silico, with only the deployment carried out on physical platforms. The core idea—SAM—is directly borrowed from machine learning literature and is already well-established.

2. Limited conceptual advancement: While the experimental implementation differs from the authors’ earlier work, the underlying idea and overall conceptual direction remain highly similar. This raises concerns about a lack of novelty in terms of scientific framing and motivation. Such conceptual continuity may be appropriate for an incremental study, but may not meet the threshold for a high-impact general-interest journal.

3. Absence of full neural network functionality: Despite being framed as work on “physical neural networks,” the experimental demonstrations are limited to simple linear functions. No physical implementation of nonlinear activation or full inference capability is presented.

4. Presentation issues: The manuscript contains several typographical issues and instances of repeated text. A careful proofreading is necessary to ensure clarity and professionalism in presentation.

That said, I appreciate the authors’ effort in generating high-quality figures and conducting thorough, cross-platform experimental validation. These aspects reflect a strong technical execution.

In summary, while the paper demonstrates a useful application of an existing training technique to physical systems, the work currently does not meet Nature Communications’ expectations for innovation and conceptual significance. Substantial improvements in both novelty and experimental depth would be required for this work to be considered for publication.

Version 1:

Reviewer comments:

Reviewer #1

(Remarks to the Author)

I have reviewed the responses the authors gave to my questions and suggestions, and the revisions they have made to the manuscript as a result. They have satisfactorily responded in all cases, so I recommend publication.

(Remarks on code availability)

香港中文大學
The Chinese University of Hong Kong

Dear Reviewers,

Thank you very much for taking your time to review our manuscript (ID: NCOMMS-25-26432-T) titled “Perfecting Imperfect Physical Neural Networks using Sharpness-Aware Training”. We appreciate all your comments and suggestions. We have carefully revised the manuscript and the Supplementary Notes accordingly. The modifications to the manuscript and Supplementary Notes are highlighted in blue.

Detailed point-by-point responses are provided below:

Response To Reviewer #1

Comment 1

Xu et al. propose to apply a technique from conventional machine learning, <<sharpness-aware training>> (SAT), to training physical neural networks (PNNs), and empirically observe that it works well when applied to some optical neural networks. While SAT was initially introduced to improve generalization, here the authors find that it also makes training more robust to noise/drifts in physical systems.

I believe the idea is sound and the results show that SAT could be generally useful in PNNs. Because of the broad utility of the approach, I strongly recommend publication of the manuscript in Nature Communications subject to the authors addressing the following comments.

Response 1

We sincerely appreciate the reviewer's high recognition of our work and valuable suggestions.

Comment 2

1. The paper seemingly overclaims novelty: SAT was not invented by the authors, but rather was previously reported in references [42] and [50]. The current abstract says: "...through a novel technique called Sharpness-Aware Training". But the technique is not novel! (Indeed it was reported 4 years ago, in the most prominent machine-learning conference.) It would be more appropriate for the abstract (and other parts of the paper) to describe what they have done as apply the (established; not novel) technique SAT to the training of PNNs and observe that it results in improved robustness to model imperfections and PNN noise/drifts etc.

Response 2

We thank the reviewer for pointing out this issue. We realize that we did not clearly explain the relationship between Sharpness-Aware Minimization (SAM) and Sharpness-Aware Training (SAT). The wording in the original abstract was inappropriate.

You are correct that SAT is inspired by the machine learning SAM, which was proposed to enhance generalization capabilities of AI models. Its core insight is the connection between the sharpness of the loss landscape and a model's generalization performance and leverage this connection to enhance generalization capabilities of AI models. Our work is inspired by this conceptual framework, however the originality of our contribution is to

establish a new connection—between the sharpness of the loss landscape and the robustness of physical systems. Guided by this new perspective, we propose SAT tailored specifically for physical systems, which enables robustness to hardware imperfections, environmental disturbance, and transferrable learning, for both in-silico and in-situ training. Such cross-disciplinary integration is the essence of interdisciplinary innovation.

In addition to introducing a new conceptual framework, our work also makes a significant methodological contribution. The original SAM cannot be directly applied to physical systems. SAM is designed to optimize weights and biases in AI models that are explicitly defined by precise mathematical functions. However, as we pointed out in Section 2.1 of the Principle section, the mathematical models of PNNs are often imprecise or even unknown. To address this, SAT extends from SAM in two ways: (1) it accommodates cases where the underlying model of the PNN is not explicitly known, and (2) it reformulates the standard weight optimization into a control-parameter optimization, allowing direct training of the physical system. These new developments make SAT highly generalizable and suitable for a wide range of physical systems, whether or not a precise model is available, and supports both in-silico and in-situ training scenarios.

Thank you for pointing out the inappropriate statement in the abstract. In the revised version, we have clarified the relationship between SAM and SAT as requested.

Revision 2

We clarify the relationship between SAM and SAT as requested in the **Abstract (Line 21-25)** and **Manuscript (Line 91-95)**.

“Here, we address the challenges with both in-silico and in-situ training through Sharpness-Aware Training (SAT), \textcolor{blue}{inspired by a machine learning technique Sharpness-Aware Minimization (SAM)}~\cite{foret2021iclr} that links loss landscape geometry with model generalization to enhance model generalization. In this work, we establish a new link between loss landscape geometry with robustness of physical systems, and then leverage this connection to tackle above challenges in training physical systems.”

“Furthermore, SAT develops methodologies for automatically locating flat minima in physical systems and enables direct training in physical systems.”

Comment 3

2. The standard approach to training a physical (analog) neural network for which one has a fairly accurate model is to perform training in a digital computer but to inject noise during training (sometimes called noise-aware training: doi:10.1109/TED.2021.3089987 ; review: doi:10.1002/aisy.202200029). For Figure 2j and for Figure 5c, it would be useful if the authors could add a comparison to noise-aware training (NAT)

or related standard noise-injection techniques. I also think it would be useful for physics-aware training (PAT) to be added as a point of comparison in Figure 2j.

Response 3

We thank the reviewer for raising the questions regarding Noise-Aware Training (NAT) and Physical-Aware Training (PAT).

We are aware that **NAT** is widely adopted to enhance model robustness against system imperfections [1-4]. In our original manuscript, we have compared our method with NAT in the Microring resonator (MRR)-based optical neural networks (ONNs), but presented the results in Supplementary Note 4.1 rather than in the main text. In the revised version, we have now included this comparison directly in the main manuscript (Figure 2j). We also add an additional comparison in the Mach-Zehnder Interferometer (MZI)-based ONNs (Figure 5c), where the results also clearly demonstrate the superior performance of SAT over NAT.

NAT enhances robustness by injecting synthetic noise into the model during training, typically assuming simple additive Gaussian distributions. However, imperfections in physical systems often deviate significantly from such simplified Gaussian assumptions. These imperfections may include modeling errors and implementation-related variations such as pixel-level misalignment, fabrication errors, and temperature-induced shifts, which are non-Gaussian. As a result, NAT cannot fully capture the complex, system-specific error characteristics and thus has limitations in addressing robustness issues arising from these non-Gaussian imperfections.

In contrast, SAT makes no prior assumptions about the nature of system imperfections. Instead, it defines a clear optimization objective (i.e., sharpness minimization) that is generally applicable across different systems. By linking system robustness to the fundamental geometry of the loss landscape, SAT inherently accounts for a wide range of imperfections, including modeling inaccuracies, environmental drift, and hardware noise, without requiring explicit modeling of each. This enables SAT to optimize PNNs more accurately and robustly, even when the exact form of imperfections is unknown.

To prove the advantage, we first train the MRR-based PNNs with NAT and SAT respectively. The training model is obtained at the room temperature of 22°C. When training with NAT, a weight noise, which follows a gaussian distribution with mean value at 0, and standard deviation at 0.3, is injected. The reason for choosing this level noise is to maximally increase the robustness while without decreasing the accuracy. At the inference stage, the TEC is removed and the room temperature becomes 20°C. Despite the temperature change being as large as 2°C, SAT maintains a high accuracy of 91.0% while the accuracy of NAT is reduced to 61.0%, as shown in Figure R1. The result now is included in the directly in the Figure 2j of main manuscript. Another evidence that SAT

outperforms NAT that the sharpness metric λ_{max} of SAT is only 1.2 while that of NAT is 414.5 (Figure R2). A larger λ_{max} indicates higher sensitivity to parameter perturbations.

Figure R1. Inference accuracy on microring circuits and confusion matrix when removing the thermoelectric controller (TEC).

Figure R2. Loss landscape and sharpness metric λ_{max} of the neural network trained by different methods.

Similar comparison is also conducted on MZI-based ONN. During training with NAT, the noise we add to the phase parameter follows a Gaussian distribution with mean value at 0, and standard deviation at 0.01 rad. The aim is to maximally increase the robustness while without decreasing the accuracy. The learning rates are set at 0.1 for both NAT and SAT, and the training epoch numbers are 40 for both NAT and SAT. After training, we evaluate the robustness by adding the phase error and beam splitter error simultaneously at 0.15 rad. And the accuracy for NAT is 79.8% compared to SAT at 97.1%. This additional result is added in Figure 5c in the revised manuscript and shown in the response letter Figure R3.

Figure R3. Training result performance comparison.

PAT is an in-situ training method designed to enable backpropagation directly on physical systems. However, it does not address resilience issues during or after training. Like other existing in-situ training approaches, PAT suffers from two key limitations: (1) The trained system is vulnerable to time-dependent degradation caused by factors such as noise, environmental disturbances, and hardware variability. (2) The trained parameters are not transferable to other systems due to device variations caused by fabrication errors.

In contrast, SAT addresses both of these limitations, and SAT can be implemented with in-situ training setting. As shown in our revised manuscript of ‘Section 2.3 In-situ SAT: facilitating transferable in-situ training’ (‘Section 2.4 Facilitate transferable in-situ training’ in the original manuscript), we compared SAT with PAT and evaluated the sensitivity of the trained models on MZI-based ONNs using the maximum eigenvalue (λ_{max}) of the Hessian matrix. A larger λ_{max} indicates higher sensitivity to parameter perturbations, and thus lower robustness. SAT exhibits significantly lower λ_{max} compared to PAT (PAT: 114.7, in-situ SAT: 0.6), indicating that models trained with SAT are much more resilient to parameter variations and system disturbances.

This is further supported by the new inference results (Figure R1) in the MRR-based ONNs as requested by the reviewer, which demonstrate that PAT fails to maintain performance under ambient temperature fluctuations when the thermoelectric controller (TEC) is removed and without retraining. In contrast, SAT maintains consistent robustness under such variations. To be more specific, a typical PAT training contains a real physical forward model and an approximate digital backward model. The forward physical model considers fabrication variance, while during the backward period, the backward digital model does not contain the fabrication variance. In this circumstance, the accuracy converges to 96.0% during training. But when removing the thermoelectric controller (TEC), the accuracy drops significantly to 59.0%. The result indicates that PAT fails to maintain performance under ambient temperature fluctuations.

Revision 3

1. We **add** the NAT trained neural network loss landscape to **Figure 2g**, remove the NAT training results from the Supplementary Note 4.1 to Figure 2j, and explain the reason why SAT outperforms NAT (**Line 181-291**).
 2. We **add** the NAT training results in Mach-Zehnder interferometer (MZI) mesh to **Figure 5c**, and include the simulation details into the manuscript (**Line 336-341**).
-

Comment 4

3. The authors have not provided the code to replicate their results. Especially for a methodology paper, this is unacceptable and against Springer Nature's stated policy on code availability. The authors should provide the code in a GitHub (or similar) repository. At the very least, this repository should contain everything needed to rerun the simulations reported in Figure 5.

Response 4

We thank the reviewer for pointing out the issue regarding code availability. Therefore, we have summarized all the simulation codes of different computing systems and store them in the **GitHub repository**.

<https://github.com/cuhkuangslab/Sharpness-Aware-Training/tree/main>

References

- [1] Kariyappa, S., Tsai, H., Spoon, K., Ambrogio, S., Narayanan, P., Mackin, C., ... & Burr, G. W. (2021). Noise-resilient DNN: Tolerating noise in PCM-based AI accelerators via noise-aware training. *IEEE Transactions on Electron Devices*, 68(9), 4356-4362.
- [2] Yang, X., Wu, C., Li, M., & Chen, Y. (2022). Tolerating noise effects in processing-in-memory systems for neural networks: a hardware - software codesign perspective. *Advanced Intelligent Systems*, 4(8), 2200029.
- [3] Wu, C., Yang, X., Yu, H., Peng, R., Takeuchi, I., Chen, Y., & Li, M. (2022). Harnessing optoelectronic noises in a photonic generative network. *Science advances*, 8(3), eabm2956.
- [4] Mourgias-Alexandris, G., Moralis-Pegios, M., Tsakyridis, A., Simos, S., Dabos, G., Totovic, A., ... & Pleros, N. (2022). Noise-resilient and high-speed deep learning with coherent silicon photonics. *Nature communications*, 13(1), 5572.

Response To Reviewer #2

Comment 1

The article introduces a novel training approach, Sharpness-Aware Training (SAT), aimed at enhancing both the training process and deployment robustness of physical neural networks (PNNs). I appreciate the insight that the authors establish a connection between sharp minima in the loss landscape and the poor robustness observed in physical systems. I believe this work has the potential to become a foundational methodology for training PNNs—specifically, by incorporating a sensitivity term of the objective function into the loss formulation to promote the learning of more robust parameters. In addition, I have a few questions and suggestions that I hope the authors can clarify, which I believe would further improve the quality of the paper.

Response 1

We sincerely thank the reviewer for the high recognition of our work. Your comments will be invaluable for improving our manuscript.

Comment 2

1. When the influence of certain parameters on the physical process is not explicitly known, approximating the gradient via finite differences offers an effective way to bypass the need for explicit modeling. Figure 3(e) illustrates the separate optimization processes for rotation, shift, and scale parameters. Have the authors considered jointly optimizing these parameters instead of optimizing them independently?

Response 2

We thank the reviewer for raising the question regarding joint parameter optimization.

In the original manuscript, we performed optimization for each parameter independently and evaluated its individual impact on performance. However, our method is also applicable to joint optimization of multiple parameters. To address the reviewer's request, we propose a sequential joint optimization strategy: optimizing the rotation angle in epoch 1, the pixel shift in epoch 2, and the scale factor in epoch 3. This approach enables joint consideration of all parameters while maintaining the same computational complexity as optimizing a single parameter at a time.

Now, we have included the joint optimization result in the manuscript (Line 448-456) and the Supplementary Note 6.3.

To be more specific, we first take rotation angle, shift pixel and scale factor as the joint optimization parameters. After training, we evaluate the model’s robustness through a 2D sweep over rotation angle and shift pixel and keep the scale factor at 1. As shown in Figure R4, our method maintains high robustness against both rotation angle and shift pixel changes after the joint optimization. Next, the robustness of the model trained with joint parameter optimization is compared with that of models trained with single-parameter optimization for rotation angle, shift pixel, and scale factor, respectively. As shown in Figure R5, joint optimization achieves comparable robustness to that of single-parameter optimization. This result demonstrates that our method is applicable to joint optimization of multiple parameters without degrading the robustness of the trained model.

Figure R4. 2D accuracy sweep with different rotation angle and shift pixel.

Figure R5. Inference accuracy with different rotation angle, shift pixel and scaling number.

Revision 2

We include the joint optimization result in the manuscript (**Line 454-462**) and the **Supplementary Note 6.3**.

Comment 3

2.Lines 390–391 state: “During the experiment, we use the affine function to quantitatively adjust rotation, shift, and scaling parameters.” Is it truly feasible to achieve such precise control over angle and pixel-level shifts in a physical experimental setup? Figure 3e appears to show results from a real experiment, but in practice, these deviations—especially angular or spatial misalignments—are typically not precisely known or controlled.

Response 3

You are correct that such deviations are not controllable in real-world implementations. However, the purpose of adjusting these parameters is not to control them, but rather to quantitatively evaluate the tolerance of our method to these variations.

During SAT training, we assume perfect alignment between the OLED and SLM. This simplification is intentional, as SAT trained systems are expected to be inherently robust to implementation errors, even when those errors are unknown. As a result, we expect the system trained by SAT to maintain high accuracy in the presence of practical misalignments such as rotation, shift, or scaling.

To quantitatively assess the tolerance of our method to such misalignment errors at the implementation stage, we follow the approach in [1] and apply affine transformations to introduce controllable deviations. We then evaluate the system's accuracy under these conditions. This procedure enables us to systematically assess the resilience of the trained model to realistic imperfections in the physical setup.

Revision 3

We revise the manuscript to illustrate our experimental implementation process with more details (**Line 416-425**).

Comment 4

3.Line 297 states that “the system trained with standard BP degrades significantly from 80.0% to 7.0% as the temperature varies from 21°C to 23°C.” However, I could not clearly observe this drastic performance drop in Figure 2(h). Could the authors clarify this discrepancy or improve the clarity of the visualization?

Response 4

We thank the reviewer for pointing out this issue. The confusion was caused by a typo. The accuracy of standard BP degrades from 80.0% to **17.0%**, not **7.0%**. We apologize for the confusion.

Revision 4

1. Correct the typo 7.0% to 17.0%.
2. We revise the **Figure 2(h)** and add the grid on the Figure 2(h) to improve the clarity of the visualization.

[Original Figure 2h]

[Revised Figure 2h]

Comment 5

4.I am curious about the tolerance limits of the proposed method. Specifically, how large can the deviations (e.g., phase errors, temperature shifts) be before the performance of the SAT-trained model degrades significantly or fails altogether?

Response 5

We thank the reviewer for raising the question regarding the tolerance limits.

The tolerance limitation is presented in Figure R6. For temperature tolerance, our experimental results show that when the temperature variation is within $\pm 1^\circ\text{C}$, the accuracy on the MNIST dataset can be maintained at around 95.0% (idea performance is 98.0%). When the temperature variation further increases, the accuracy drops significantly. For more complex tasks, such as image reconstruction and generation, the system becomes more sensitive to imperfections. In these cases, performance metrics such as Mean square error (MSE) and Fréchet Inception Distance (FID) remain stable at around 0.023 (Ideal MSE is 0.013) and 120.59 (Ideal FID is 100.31) within a temperature fluctuation of $\pm 0.25^\circ\text{C}$. Beyond this threshold, for example, when the temperature variation reaches $+0.5^\circ\text{C}$, the image quality metrics MSE and FID degrades to 0.547 and 426.69, respectively.

In our results, we select $\pm 0.25^\circ\text{C}$ as the thermal fluctuation limit, as it represents the maximum expected fluctuation in current co-packaged optical modules [2].

Regarding phase errors, we set both the standard deviation of phase errors (σ_{ps}) and the beam splitter errors (σ_{bs}) to 0.15 rad. These values correspond to typical wafer-scale variations observed in current standard manufacturing processes [3]. Our results show that under such conditions, SAT can maintain an accuracy of 95.2% (the ideal performance is 97.1%), whereas without SAT, the accuracy drops to 58.6%. When the errors are further increased to 0.20 rad, even with SAT, the performance degrades to 61.3%.

Figure R6. Inference accuracy with different levels of temperature change and phase errors.

Comment 6

5.The method effectively mitigates performance degradation caused by various noise sources, but how is the trade-off between sharpness and loss handled in practice? How should the hyperparameter α be chosen? A similar issue arises with the self-defined hyperparameters α_1 , r , μ , and ρ . It would be beneficial if the authors could provide a principled guideline or sensitivity analysis.

Response 6

We thank the reviewer for raising the question about the trade-off between sharpness and loss, as well as the principled guideline.

After carefully reading the manuscript and Supplementary Notes, we find some defined hyperparameters are redundant. Now we have simplified all definitions, and we apologize for any possible confusions. The conclusion is, parameters defined in the initial manuscript and Supplementary Note α_1 , r , μ , and ρ , are not necessary. The only required hyperparameter is α . Moreover, The value of the hyperparameter α determines the penalty of sharpness for loss during the actual training process, that is, the trade-off between sharpness and loss.

In the following response, we will first introduce how to simplify the redundant hyperparameters to only α , and introduce our principled guideline for selecting the α value through sensitivity analysis.

The hyperparameters α , α_1 , r were noted in the section 2.1 of the original maintext, and the hyperparameters μ , and ρ were noted in section 2.3. Here, we show how to simplify these hyperparameters step by step.

In section 2.1, we initially defined the loss function \mathcal{L}_1 as (Equation (2) in the original main text),

$$\mathcal{L}_1 = \mathcal{L}(y, y_{target}; \Theta) + \alpha \left\| \frac{\partial \mathcal{L}(y, y_{target}; \Theta)}{\partial \Theta} \right\|_2 \quad (R1)$$

The hyperparameter α indicates how large the penalty of the regularization term should be added to the initial loss function.

To simplify the computational complexity, we applied the method in [4] by approximating the Hessian matrix through first-order Taylor expansion, as shown in Equation (R2-R6) (The original Equation (S2-S6) in Supplementary Note 1.1).

$$\frac{\partial \left\| \frac{\partial \mathcal{L}(y, y_{target}; \Theta)}{\partial \Theta} \right\|_2}{\partial \Theta} = \frac{\partial^2 \mathcal{L}(\Theta)}{\partial \Theta^2} \cdot \frac{\partial \mathcal{L} / \partial \Theta}{\| \partial \mathcal{L} / \partial \Theta \|_2} \quad (R2)$$

$$\mathcal{L}(\Theta + \Delta\Theta) \approx \mathcal{L}(\Theta) + \frac{\partial \mathcal{L}(\Theta)}{\partial \Theta} \cdot \Delta\Theta \quad (R3)$$

$$\frac{\partial \mathcal{L}(\Theta + \Delta\Theta)}{\partial \Theta} \approx \frac{\partial \mathcal{L}(\Theta)}{\partial \Theta} + \frac{\partial^2 \mathcal{L}(\Theta)}{\partial \Theta^2} \cdot \Delta\Theta \quad (R4)$$

$$\frac{\partial \left\| \frac{\partial \mathcal{L}(y, y_{target}; \Theta)}{\partial \Theta} \right\|_2}{\partial \Theta} \approx \frac{1}{r} \left(\frac{\partial \mathcal{L}(\Theta + \Delta\Theta)}{\partial \Theta} - \frac{\partial \mathcal{L}(\Theta)}{\partial \Theta} \right) \quad (R5)$$

$$\frac{\partial \mathcal{L}_1(\Theta)}{\partial \Theta} = \frac{\partial \mathcal{L}(\Theta)}{\partial \Theta} + \frac{\alpha}{r} \left(\frac{\partial \mathcal{L}(\Theta + \Delta\Theta)}{\partial \Theta} - \frac{\partial \mathcal{L}(\Theta)}{\partial \Theta} \right) \Big|_{\Delta\Theta=r \frac{\partial \mathcal{L}/\partial \Theta}{\|\partial \mathcal{L}/\partial \Theta\|_2}} = \frac{\partial \mathcal{L}(\Theta)}{\partial \Theta} + \alpha_1 \left(\frac{\partial \mathcal{L}(\Theta + \Delta\Theta)}{\partial \Theta} - \frac{\partial \mathcal{L}(\Theta)}{\partial \Theta} \right) \Big|_{\Delta\Theta=r \frac{\partial \mathcal{L}/\partial \Theta}{\|\partial \mathcal{L}/\partial \Theta\|_2}} \quad (R6)$$

During the above calculation, the new hyperparameters r and α_1 were introduced. The hyperparameter r indicates how large the perturbation $\Delta\Theta$ should be given to Θ .

We directly set $r = \alpha$ during training all the ONNs and it did not influence the final training results. In this circumstance, α_1 equals 1. Therefore, the above 3 hyperparameters α , α_1 , r are simplified into only α . And Equation (R6) is simplified into Equation (R7),

$$\frac{\partial \mathcal{L}_1(\Theta)}{\partial \Theta} = \frac{\partial \mathcal{L}(\Theta)}{\partial \Theta} + \frac{\alpha}{\alpha} \left(\frac{\partial \mathcal{L}(\Theta + \Delta\Theta)}{\partial \Theta} - \frac{\partial \mathcal{L}(\Theta)}{\partial \Theta} \right) \Big|_{\Delta\Theta=\alpha \frac{\partial \mathcal{L}/\partial \Theta}{\|\partial \mathcal{L}/\partial \Theta\|_2}} = \frac{\partial \mathcal{L}(\Theta + \Delta\Theta)}{\partial \Theta} \Big|_{\Delta\Theta=\alpha \frac{\partial \mathcal{L}/\partial \Theta}{\|\partial \mathcal{L}/\partial \Theta\|_2}} \quad (R7)$$

Correspondingly, Equation (3) and (4) (Now Equation (3) in the main text) are simplified with only α , as shown in Equation (R8).

$$\Delta\Theta = \alpha \frac{\partial \mathcal{L}/\partial \Theta}{\|\partial \mathcal{L}/\partial \Theta\|_2}$$

$$\frac{\partial \mathcal{L}_1}{\partial \Theta} = \frac{\partial \mathcal{L}(\Theta + \Delta\Theta)}{\partial \Theta} \Big|_{\Delta\Theta=\alpha \frac{\partial \mathcal{L}/\partial \Theta}{\|\partial \mathcal{L}/\partial \Theta\|_2}} \quad (R8)$$

In section 2.3, we originally defined two new hyperparameters, μ and ρ . The meaning of these two hyperparameters is still how large the perturbation should be given to the corresponding parameters, rotation angle θ and weights W . Initially we used two hyperparameters, μ and ρ to indicate that their values can be selected independently.

Now we have changed the hyperparameters defined in Equation (5) and (6) (Now Equation (4) in the main text) in the main text to α_1 and α_2 (originally was μ and ρ), as shown in Equation (R9).

$$\theta_{adv} = \theta_0 + \alpha_1 \frac{d\mathcal{L}/d\theta}{\|d\mathcal{L}/d\theta\|_2}$$

$$W_{adv} = W_t + \alpha_2 \frac{\nabla_W \mathcal{L}(W_t)}{\|\nabla_W \mathcal{L}(W_t)\|_2} \quad (R9)$$

Here α_1 and α_2 determine the perturbations added to the rotation angle and weights, respectively. The function of α_1 and α_2 are equivalent to α in Equation (R1). Because parameters Θ and weights W are optimized separately, we use different subscript to indicate the difference.

Above all, all previously defined hyperparameters can be simplified into a single hyperparameter α , which defines how large the perturbation $\Delta\Theta$ should be given.

Next, we introduce our principled guideline for selecting the α value through sensitivity analysis. To be more specific, we perform sensitivity analysis on both the robustness and accuracy of the trained model. Here, we choose the Microring resonator (MRR)-based computing system as an example demonstrating system. We use the sensitivity λ_{max} to evaluate the trained model’s robustness. Small λ_{max} indicates high system robustness.

The sensitivity analysis result is shown in Figure R5, by gradually increasing the α from 0 to 100, the model’s sensitivity first reduces and then increases. In contrast, the model’s accuracy remains stable when α is smaller than 1, and drops significantly when we further increase the value of α . This result indicates that small perturbations can effectively increase the model’s robustness while maintaining the accuracy, but larger perturbations (Large sharpness penalty) would make the training totally ineffective.

Based on the sensitivity analysis result, we choose the optimum value of α at 0.1, because of the highest model’s robustness and accuracy are achieved simultaneously at this point.

Figure R7. Sensitivity analysis

Revision 6

We simplify the definition of hyperparameters in **Equation 2-4** and give a detailed principled guideline of choosing the hyperparameter in the manuscript (Line 154-155, Line 162-166) and **Supplementary Note 1.1**.

Comment 7

6.Minor issues: In Figure 2(h), it would be more accurate to describe the performance shown as the ideal performance rather than the theoretical limit. In Equation (4) of the main text, the numerator appears to be missing the letter "L".

Response 7

We thank the reviewer for pointing out the inexact wording in our expressions and typo. We have revised the manuscript and corrected the errors.

Revision 7

1. Revise ‘theoretical limit’ to ‘ideal performance’.
2. Revise **Equation (4)** (Now Equation (3) in the main text).

Comment 8

7.I strongly encourage the authors to consider releasing the code, even if only for the MZI-based implementation, which would significantly enhance the impact and reproducibility of this work.

Response 8

We thank the reviewer for pointing out the issue regarding code availability. Therefore, we have summarized all the simulation codes of different computing systems and store them in the **GitHub repository**.

<https://github.com/cuhkuangslab/Sharpness-Aware-Training/tree/main>

Reference

- [1] Zheng, Z., Duan, Z., Chen, H., Yang, R., Gao, S., Zhang, H., ... & Lin, X. (2023). Dual adaptive training of photonic neural networks. *Nature Machine Intelligence*, 5(10), 1119-1129.
- [2] Wu, S., Wen, S., & Xue, H. (2024, August). Liquid-Cooled Heat Dissipation Technology for Co-Packaged Optics over 12.8 Tbps. In *2024 25th International Conference on Electronic Packaging Technology (ICEPT)* (pp. 1-4). IEEE.

[3] Bandyopadhyay, S., Hamerly, R., & Englund, D. (2021). Hardware error correction for programmable photonics. *Optica*, 8(10), 1247-1255.

[4] Foret, P., Kleiner, A., Mobahi, H., & Neyshabur, B. (2020). Sharpness-aware minimization for efficiently improving generalization. arXiv preprint arXiv:2010.01412.

Response To Reviewer #3

Comment 1

This manuscript introduces the use of Sharpness-Aware Minimization (SAM), a technique originally developed in the machine learning domain, to improve the robustness of physical neural networks. By simultaneously minimizing the loss and its sharpness, the authors aim to eliminate the need for in-situ training, thereby enabling the network to tolerate environmental fluctuations and fabrication variance. The effectiveness of this approach is experimentally evaluated across three distinct hardware platforms.

Response 1

We thank the reviewer for the comments, which help us improve our manuscript.

Comment 2

While the manuscript presents a technically sound implementation and provides extensive experimental validation, I have the following major concerns regarding its novelty and overall impact:

1. Lack of originality in methodology: The training is entirely performed in silico, with only the deployment carried out on physical platforms. The core idea—SAM—is directly borrowed from machine learning literature and is already well-established.

Response 2

We thank the reviewer for their comments. However, we respectfully disagree with the points raised regarding the in silico training. The reviewer comments that “the training is entirely performed in silico, with only the deployment carried out on physical platforms”. However, our training method, SAT, is applicable to both in-silico and in-situ training. In particular, the whole **Subsection 2.3 (originally Subsection 2.4)** is dedicated to reporting the results of applying SAT in an in-situ training setting, demonstrating its unique ability to enable *transferable learning*, that is, parameters trained in one device can be successfully transferred to other devices. In contrast, existing methods such as DAT [1] and PAT [2] lack this capability and are limited by device-specific training constraints.

Regarding the second comment “The core idea—SAM—is directly borrowed from machine learning literature and is already well-established.”, we would like to clarify that that drawing inspiration from machine learning should not be the reason to diminish the research novelty.

SAM was originally proposed in the machine learning community. Its core insight is the connection between the sharpness of the loss landscape and a model's generalization performance. The purpose is to address AI model generalization. Our work is inspired by this conceptual framework, however the originality of our contribution is to establish a new connection—between the sharpness of the loss landscape and the robustness of physical systems. Guided by this new perspective, we propose SAT tailored specifically for physical systems, which enables robustness to hardware imperfections, environmental disturbance, and transferrable learning, for both in-silico and in-situ training. Such cross-disciplinary integration is the essence of interdisciplinary innovation.

In addition to introducing a new conceptual framework, our work also makes a significant methodological contribution. The original SAM cannot be directly applied to physical systems. SAM is designed to optimize weights and biases in AI models that are explicitly defined by precise mathematical functions. However, as we point out in Section 2.1 of the Principle section, the mathematical models of PNNs are often imprecise or even unknown. To address this, SAT extends from SAM in two ways: (1) it accommodates cases where the underlying model of the PNN is not explicitly known, and (2) it reformulates the standard weight optimization into a control-parameter optimization problem, allowing direct training of the physical system, thus enabling in-situ training. These new developments make SAT highly generalizable and suitable for a wide range of physical systems, whether or not a precise model is available, and support both in-silico and in-situ training scenarios.

To support the conceptual and methodological novelty and effectiveness of our approach, we have conducted experiments and simulations on three different physical systems: microring circuits, free-space optics, and MZI networks. These systems were not chosen to simply repeat the same concept, but rather to demonstrate the broad applicability and benefits of SAT in different contexts below:

- (1) When SAT is implemented in in-silico training setting, it addresses the modeling challenges in in-silico training by enabling high-performance training even when the digital model is imperfect;
- (2) When SAT is implemented in in-situ training setting, it facilitates transferable training, overcoming a key limitation of in-situ training where trained parameters typically cannot be transferred to other systems due to hardware variability;
- (3) SAT enables high-performance inference when environmental disturbances and system noise presents after training regardless of using in-silico or in-situ training; and
- (4) it is generalizable across a wide range of physical systems, regardless of whether a precise mathematical model is available.

We realize that some key contributions of our work were not clearly visualized in the original manuscript. In response, we have significantly revised the manuscript to better highlight the novelty and significance of our contributions by adding an additional table (Table 1) in the between Section 3 and 4, and additional paragraph at the end of Section 2.1.

We sincerely hope that the reviewer will reconsider the evaluation after reading the above clarifications and improvements.

Table R1. Comparison of training methods regarding training objectives, generality, and performance during the training, deployment, and inference stages.

Table 1: Comparison of training methods regarding training objectives, generality, and performance during the training, deployment, and inference stages.

Training Method	Training Objective	Generality	Training Stage			Deployment Stage	Inference stage	
			Tolerance to model-reality gap	Training Speed	Robustness		Scalability	Accuracy (under ambient perturbations)
In-silico training	Loss	High ✓	Low	$O(T_0)$ ✓	Low	High ✓	Low	Mid
Gradient approx. by data driven (PAT [5], DAT [29])		High ✓	Mid	$O(T_0)$ ✓		High ✓		
In-situ training	Loss	Limited (Require symmetry)	High ✓	$>O(T_0)$	Low	Mid	High ✓	Low
Gradient estimation (Finite difference [33–35])			High ✓	$O(T_0)$ ✓		High ✓		
Gradient measurement (Fully forward [36], Adjoint [37, 39])			High ✓	$>O(T_0)$		Mid		
Physical local learning (Forward forward [4])			High ✓	$>O(T_0)$		Mid		
Direct feedback alignment [42, 43]			High ✓	$>O(T_0)$		Mid		
Gradient free methods (GA, SO, ES)			High ✓	$>>O(T_0)$		Mid		
In-silico NAT	Loss	High ✓	Mid (Gaussian errors only)	$O(T_0)$ ✓	Mid (Gaussian errors only)	High ✓	Mid	Mid
In-situ NAT						High ✓	Mid	Mid
In-silico SAT (Ours)	Loss & Sharpness ✓	High ✓	High ✓	$O(T_0)$ ✓	Highest ✓	High ✓	High ✓	High ✓
In-situ SAT (Ours)			Highest ✓		High ✓	Highest ✓	Highest ✓	Highest ✓

1. T_0 is the time to convergence for backpropagation [55]. PAT-Physical aware training. DAT-Dual adaptive training. GA-Genetic algorithm. SO-Surrogate optimization. ES-Evolutionary strategy. NAT-Noise aware training. SAT-Sharpness aware training.
2. Generality means whether the method is general to different physical systems.
3. Scalability means whether the method is still effective when the physical system and task complexity scale up.

Revision 2

We have significantly revised the manuscript to better highlight the novelty and significance of our contributions by adding an additional table (Table 1) in the between Section 3 and 4, and additional paragraph at the end of Section 2.1 (Line 190-201). We clarify the relationship between SAM and SAT as requested in the Abstract (Line 21-25) and Manuscript (Line 91-95).

“In the following section, we demonstrate SAT on three distinct PNNs. These demonstrations not only highlight the broad applicability of SAT, but also illustrate its advantages over existing training methods from three perspectives: (1) When SAT is implemented in in-silico training setting, it closes the model-reality gap, enabling accurate backpropagation training under imprecise models (Section.\ref{sec2.2}); (2) When SAT is implemented in in-situ training setting, SAT addresses the challenge of limited transferability caused by fabrication variances. Parameters trained on one device using SAT can be reliably deployed to other devices without accuracy degradation, even when hardware discrepancies exist (Section.\ref{sec2.3}); (3) In both in-silico and in-situ training settings, PNNs trained by SAT can continuously operate accurately after training under perturbations without retraining (Section.\ref{sec2.2}--Section.\ref{sec2.4}); (4) SAT is broadly applicable to physical systems regardless of whether an accurate physical model is explicitly available (Section.\ref{sec2.2}--\ref{sec2.4}). The advantages of SAT over existing in-silico and in-situ training methods, including those designed to improve

robustness, such as NAT and optical pruning, are further demonstrated in the following sections, and summarized in Table.\ref{tab:training_comparison}.”

“Here, we address the challenges with both in-silico and in-situ training through Sharpness-Aware Training (SAT), \textcolor{blue}{inspired by a machine learning technique Sharpness-Aware Minimization (SAM)\cite{foret2021iclr}} that links loss landscape geometry with model generalization to enhance model generalization. In this work, we establish a new link between loss landscape geometry with robustness of physical systems, and then leverage this connection to tackle above challenges in training physical systems.”

“Furthermore, SAT develops methodologies for automatically locating flat minima in physical systems and enables direct training in physical systems.”

Comment 3

2. Limited conceptual advancement: While the experimental implementation differs from the authors’ earlier work, the underlying idea and overall conceptual direction remain highly similar. This raises concerns about a lack of novelty in terms of scientific framing and motivation. Such conceptual continuity may be appropriate for an incremental study, but may not meet the threshold for a high-impact general-interest journal.

Response 3

We thank the reviewer for raising the question regarding the comparison with our prior work.

In our prior pruning work [3], we also aimed to train the system to a robust region. However, the method we employed was limited to relatively simple systems where such robust regions could be easily identified. For instance, in microring circuits, where each component operates almost independently, the robust region corresponds to the flat region of each individual microring's transfer function.

However, in more general physical systems, robust regions are not so easily defined. For more complex systems such as MZI networks (Section 2.3), although their behavior can be approximated by mathematical models, the robust region of the entire network is neither explicit nor easily identifiable. In even more complex scenarios, such as free-space optical systems (Section 2.4), even the system model cannot be explicitly formulated at all, making it practically impossible to identify robust regions using the approach from our previous work. Moreover,

through this work, we found that even in simple systems, the robust region of individual components does not necessarily overlap with the robust region of the system as a whole.

In contrast, the core contribution of SAT is that it is broadly applicable to all physical systems: SAT can automatically identify robust minima without requiring explicit knowledge of the underlying physical dynamics. Our experimental and simulation results on different systems support this generality claim. Furthermore, as illustrated in Figure 1(d) in the main text, SAT not only enhances the stability of individual components but also identifies configurations that improve the robustness of the system as a whole, thus leading to superior performance even when operating on the same hardware.

Figure R8 shows the improved performance of SAT on microring circuits compared to our previous optical pruning method. As mentioned earlier, the key reason is that SAT not only enhances the stability of individual components but also identifies configurations that improve the robustness of the system as a whole. In contrast, the optical pruning approach focuses solely on stabilizing individual components.

Figure R8. Inference accuracy on microring circuits and confusion matrix when removing the thermoelectric controller (TEC).

Revision 3

We revise our manuscript to highlight the difference of our proposed method with our prior work (**Line 174-181**).

“The loss minimization process highlights the difference with our prior work optical pruning [27]. Optical pruning focuses solely on improving the stability of individual devices inside the PNNs. In contrast, SAT not only improves the stability of individual components but also enhances the overall system's stability by ensuring that the entire system's loss function remains minimized in response to weight perturbations. Furthermore, optical pruning is limited to relatively simple systems where such stable regions could be easily identified. For instance, in microring circuits, where each component operates almost independently, the robust region corresponds to the flat region of each individual microring's transfer function. In contrast, SAT is broadly applicable to different physical systems because it can automatically identify robust minima without requiring explicit knowledge of the underlying physical dynamics.”

Comment 4

3. Absence of full neural network functionality: Despite being framed as work on “physical neural networks,” the experimental demonstrations are limited to simple linear functions. No physical implementation of nonlinear activation or full inference capability is presented.

Response 4

We thank the reviewer for raising the question of the full neural network functionality.

The principle described in Section 2.1 demonstrates that SAT is universally applicable to systems with or without nonlinear functions. SAT identifies flat minima based on gradients, which can be approximated from an imperfect model or measured using finite-difference methods. Therefore, even for systems with nonlinearities, the gradient approximation process is the same as for systems with linear functions.

To further address the reviewer’s concern, we have included an additional simulation study on a deep diffractive neural network (D2NN). Here we follow the setting in [1] and include nonlinear functions between the linear diffractive neural network, as illustrated in Figure R9. The simulated system consists of two programmable linear layers and two nonlinear layers. The first nonlinear layer is implemented using a saturable absorber, which introduces an intensity-dependent transmission while preserving the optical phase [4]. And the second nonlinear component is the photodetector (PD) array, which introduces a quadratic nonlinearity during the optical-to-electrical conversion process. The input image is encoded via a spatial light modulator (SLM), after which light propagates through the layered structure (linear \rightarrow nonlinear \rightarrow linear \rightarrow nonlinear) and is ultimately captured by the PD array. We apply this system to perform handwritten digits classification, where the output class is determined by the index of the maximum PD response.

We train the physical network using both standard backpropagation (BP) and our SAT. During training, we employ the finite difference method to approximate the gradient of the loss with respect to the rotation angle (which acts as trainable control parameters), and we intentionally introduce perturbations in rotation to evaluate robustness.

As shown in Figure R10, our method maintains high classification accuracy under rotation angle variations. These results clearly demonstrate that SAT remains effective when both linear and nonlinear operations are physically implemented. The detailed simulation parameters are depicted in Table.R1. The simulation code will be provided in the open-source code file [5].

Table.R2 Simulation hyperparameters

Parameter name	Value
Batch size	200
Learning rate	0.002
Linear layers size	128*128
Linear layer pixel resolution	20 μm
Laser wavelength	532 nm

Figure R9. Schematic diagram of the simulated Deep diffraction neural network (D2NN).

Figure R10. Inference accuracy with rotation angle change from 0.0° to 1.0° .

Revision 4

We revise the manuscript to add a discussion on our method's compatibility with nonlinear systems (**Line 463-474**), and incorporate the simulation results into the **Supplementary Note 6.4**.

Comment 5

4. Presentation issues: The manuscript contains several typographical issues and instances of repeated text. A careful proofreading is necessary to ensure clarity and professionalism in presentation.

Response 5

We thank the reviewer for pointing out the presentation issues. We have done a careful proofreading and revised the typographical issues and unintended redundancies. However, certain repetitions were intentionally retained as they serve as reminders for readers, given the paper's length and technical complexity.

Comment 6

That said, I appreciate the authors' effort in generating high-quality figures and conducting thorough, cross-platform experimental validation. These aspects reflect a strong technical execution.

In summary, while the paper demonstrates a useful application of an existing training technique to physical systems, the work currently does not meet Nature Communications' expectations for innovation and conceptual significance. Substantial improvements in both novelty and experimental depth would be required for this work to be considered for publication.

Response 6

We sincerely thank the reviewer's appreciation in our large amount of effort in experiments and simulations to proof the effectiveness and advancement of our proposed method.

However, we sincerely hope that, after re-reviewing the response letter (particularly the difference and connection between SAM and SAT in **Response 1** and reviewing the in-situ training section 2.3) and the revised manuscript, the reviewer may reconsider the assessment of the novelty and contribution of our work. To the best of our knowledge, we are the first to establish a new conceptual framework that links the sharpness of the loss landscape to the robustness of physical systems. This framework leads to a new, highly effective, and universally applicable approach to training physical neural networks, effectively addressing the fundamental challenges inherent in both in-silico and in-situ training methods, as outlined below:

1. SAT allows PNNs to achieve reliability, efficiency, and precision even when facing all kinds of errors crossing design, manufacturing, and deployment stages.

- Modeling Errors: SAT does not require precise modeling of the physical system or consideration of side effects or system imperfections, yet it still outperforms methods that do.

- **Fabrication Variances:** The SAT-trained integrated PNNs are highly resilient to fabrication variances and errors, addressing a critical challenge in scaling functional large-scale PNNs.
- **Ambient Fluctuations:** PNNs trained with SAT maintain high accuracy even under temperature fluctuations, eliminating the need for power-hungry temperature control.
- **Deployment Errors:** PNNs trained with SAT show high resilience to deployment errors, such as alignment errors commonly seen in free-space PNNs including the “x” and “y” axes, rotation, and scaling.

2. SAT outperforms state-of-the-art training methods in both in-silico and in-situ training regarding accuracy, efficiency, and robustness.

First, SAT achieves the highest accuracy while significantly reducing training time compared to the most effective training methods currently based on backpropagation (Nature 601(7894), 2022; Science 380(6643), 2023; Nature Machine Intelligence 5(10), 2023) [6-8]. Second, SAT demonstrates robustness and tolerance to system imperfections, particularly those encountered after training during deployment, such as temperature fluctuations and alignment errors. This capability positions SAT as a leading solution in the field.

3. SAT facilitates transferable learning across different devices, a feature that has not been achieved with other training methods.

In modern AI hardware, transferable learning is essential and commonly practiced for digital computers. Typically, training occurs on high-end computers or cloud-based clusters, with the trained parameters then deployed across multiple edge devices for inference.

However, in physical systems, whether through in-silico or in-situ training, transferable learning has remained elusive. This is because physical systems, even with identical designs, can exhibit different behaviors due to fabrication variances and environmental noise. In contrast, SAT enables transferable training across diverse devices regardless of these imperfections—training is done ONCE in one PNN, while trained parameters can be reliably deployed to numerous edge PNNs for inference.

4. SAT is a universally applicable method for various PNNs regardless of whether their models are explicitly known or unknown, linear or nonlinear. It is versatile across diverse applications, including image classification, compression, reconstruction and generation.

In summary, SAT offers a practical, effective, and computationally efficient solution, not only for training PNNs, but also for deploying them in real-world environments and applications. We respectfully hope that the reviewer will find the clarifications and improvements satisfactory, and reconsider the overall assessment.

Reference

- [1] Zheng, Z., Duan, Z., Chen, H., Yang, R., Gao, S., Zhang, H., ... & Lin, X. (2023). Dual adaptive training of photonic neural networks. *Nature Machine Intelligence*, 5(10), 1119-1129.
- [2] Wright, L. G., Onodera, T., Stein, M. M., Wang, T., Schachter, D. T., Hu, Z., & McMahon, P. L. (2022). Deep physical neural networks trained with backpropagation. *Nature*, 601(7894), 549-555.
- [3] Xu, T., Zhang, W., Zhang, J., Luo, Z., Xiao, Q., Wang, B., ... & Huang, C. (2024). Control-free and efficient integrated photonic neural networks via hardware-aware training and pruning. *Optica*, 11(8), 1039-1049.
- [4] Lin, X., Rivenson, Y., Yardimci, N. T., Veli, M., Luo, Y., Jarrahi, M., & Ozcan, A. (2018). All-optical machine learning using diffractive deep neural networks. *Science*, 361(6406), 1004-1008.
- [5] <https://github.com/cuhkhuangslab/Sharpness-Aware-Training/tree/main>
- [6] Wright, L. G., Onodera, T., Stein, M. M., Wang, T., Schachter, D. T., Hu, Z., & McMahon, P. L. (2022). Deep physical neural networks trained with backpropagation. *Nature*, 601(7894), 549-555.
- [7] Pai, S., Sun, Z., Hughes, T. W., Park, T., Bartlett, B., Williamson, I. A., ... & Miller, D. A. (2023). Experimentally realized in situ backpropagation for deep learning in photonic neural networks. *Science*, 380(6643), 398-404.
- [8] Zheng, Z., Duan, Z., Chen, H., Yang, R., Gao, S., Zhang, H., ... & Lin, X. (2023). Dual adaptive training of photonic neural networks. *Nature Machine Intelligence*, 5(10), 1119-1129.